# BOFormer: Learning to Solve Multi-Objective Bayesian Optimization via Non-Markovian RL

**Yu-Heng Hung**[1][†]  **Kai-Jie Lin**[1][†]  **Yu-Heng Lin**[1]  **Chien-Yi Wang**[2][‡]  **Cheng Sun**[2]  **Ping-Chun Hsieh**[1][‡]
[1]National Yang Ming Chiao Tung University, Hsinchu, Taiwan
[2]NVIDIA Research
{hungyh.cs08,pinghsieh}@nycu.edu.tw

## Abstract

Bayesian optimization (BO) offers an efficient pipeline for optimizing black-box functions with the help of a Gaussian process prior and an acquisition function (AF). Recently, in the context of single-objective BO, learning-based AFs witnessed promising empirical results given its favorable non-myopic nature. Despite this, the direct extension of these approaches to multi-objective Bayesian optimization (MOBO) suffer from the *hypervolume identifiability issue*, which results from the non-Markovian nature of MOBO problems. To tackle this, inspired by the non-Markovian RL literature and the success of Transformers in language modeling, we present a generalized deep Q-learning framework and propose *BOFormer*, which substantiates this framework for MOBO via sequence modeling. Through extensive evaluation, we demonstrate that BOFormer constantly outperforms the benchmark rule-based and learning-based algorithms in various synthetic MOBO and real-world multi-objective hyperparameter optimization problems. We have made the source code publicly available to encourage further research in this direction.[*]

## 1 Introduction

Bayesian optimization (BO) offers a sample-efficient pipeline for optimizing black-box functions in various practical applications, such as hyperparameter optimization (Lindauer et al., 2022; Snoek et al., 2012; Klein et al., 2017), analog circuit design (Lyu et al., 2018; Zhou et al., 2020), and automated scientific discovery (Ueno et al., 2016; Gómez-Bombarelli et al., 2018). Notably, these real-world engineering tasks usually involve multiple objective functions, which are potentially conflicting. To search for the set of candidate solutions under a sampling budget, *multi-objective BO* (MOBO) integrates the following two components: (i) MOBO utilizes Gaussian processes (GP) as a surrogate function prior for capturing the underlying structure of each objective function and thereby offering posterior predictive distributions in a compact manner; (ii) MOBO then iteratively determines the samples through an acquisition function (AF), which induces an index-type strategy based on the posterior distributions. The existing AFs are built on various design principles, such as maximizing one-step expected improvement (Emmerich & Klinkenberg, 2008; Yang et al., 2019) and maximizing one-step information gain (Hernández-Lobato et al., 2016; Belakaria et al., 2019; Tu et al., 2022). However, the existing AFs for MOBO are mostly handcrafted and *myopic*, *i.e.,* greedily optimize a one-step surrogate and lack long-term planning capability. With that said, one important and yet under-explored challenge of MOBO lies in the design of *non-myopic* AFs.

In single-objective BO (SOBO), one promising non-myopic approach is to recast BO as a reinforcement learning (RL) problem, and several RL-based algorithms (Volpp et al., 2020; Hsieh et al., 2021; Shmakov et al., 2023) have recently witnessed competitive empirical results. Specifically, AFs could be parameterized by neural networks and learned by either actor-critic (Volpp et al., 2020) or valued-based RL (Hsieh et al., 2021), where the state-action representation consists of the posterior mean and variance of a candidate point as well as the best function value observed so far, and the reward is defined as a function of negative simple regret, as shown in Figure 1. However, a direct

---

[†]Equal contribution.
[‡]Equal advising.
[*]https://hungyuheng.github.io/BOFormer/

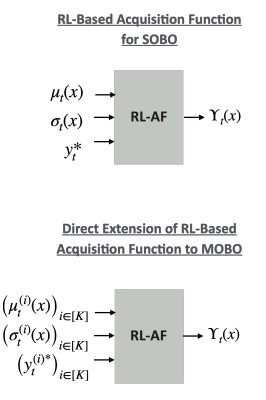 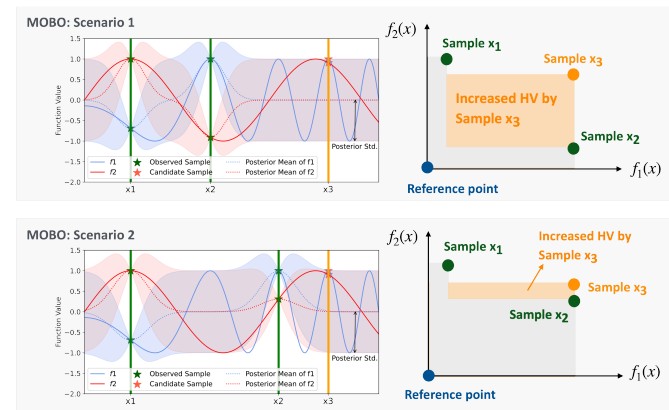

Figure 1: *Left*: In SOBO, an RL-based AF (*e.g.,* FSAF (Hsieh et al., 2021)) takes the posterior mean and standard deviation $(\mu_t(x), \sigma_t(x))$ and the best function value observed so far $y_t^*$ as input and then outputs the AF value $\Upsilon_t(x)$. An direct extension to MOBO simply takes into account the same set of information about all the $K$ objective functions. *Right*: The hypervolume identifiability issue can be illustrated by comparing the hypervolume improvement incurred by the sample $x_3$ in the two different scenarios above. Clearly, despite that the AF inputs at $x_3$ are the same in both scenarios, the increases in hypervolume upon sampling $x_3$ are rather different. Hence, the increase in hypervolume is not identifiable solely based on the AF input $(\mu_t^{(i)}(x), \sigma_t^{(i)}(x)), y_t^{(i)*})_{i \in [K]}$ of the existing RL-based AFs.

extension of these single-objective RL-based AFs to MOBO could suffer from the fundamental *hypervolume identifiability issue*. To better illustrate this, we provide a motivating example in Figure 1, which shows that one can construct a pair of scenarios that cannot be distinguished based solely on the current posterior distributions and the current best values. This example also highlights that the identifiability issue actually results from the inherent non-Markovianty in MOBO as the improvement in hypervolume is *history-dependent*. Note that this identifiability issue is much milder in the SOBO setting since a good candidate point (*i.e.,* with a high function value) remains good regardless of the history. As a result, there remains one important open challenge in MOBO:

*How to learn a non-myopic AF for MOBO without suffering from the above identifiability issue?*

To tackle the above challenge, we propose to rethink MOBO from the perspective of non-Markovian RL via sequence modeling. Motivated by the optimality equations in the general non-Markovian RL (Dong et al., 2022), we tackle the hypervolume identifiability issue by presenting the non-Markovian version of deep Q-network termed *Generalized DQN*, which extends the standard Markovian DQN (Mnih et al., 2013) by learning the generalized optimal Q-function defined on the history of observations and actions. To implement Generalized DQN, inspired by the significant success of the Transformers in language modeling, we propose BOFormer, which leverages the sequence modeling capability of the Transformer architecture and thereby minimizes the generalized temporal difference loss. As a general-purpose multi-objective optimization solver, the proposed BOFormer is trained solely on synthetic GP functions and can be deployed to optimize unseen testing functions.

Moreover, to facilitate the training process, we present several useful and practical enhancements for BOFormer: (i) *Q-augmented observation representation*: Regarding the representation of per-step observation, we propose to use the posterior of the candidate point augmented with its $Q$-value, which serves as an informative indicator of the prospective improvement in hypervolume. Under this design, the representation is completely domain-agnostic and memory-efficient in the sense that it does not increase with the domain size. (ii) *Prioritized trajectory replay buffer and off-policy learning*: To improve convergence and data efficiency during training, we utilize a prioritized trajectory replay buffer, which can be viewed as a generalization of the typical replay buffer of vanilla DQN. Through this buffer, BOFormer naturally supports off-policy learning and more flexible reuse of training data. (iii) *Demo-policy-guided exploration*: While randomized exploration (*e.g.,* epsilon-greedy) remains popular in many RL algorithms, this exploration scheme can be very inefficient as the sampling budget in BO is usually much smaller than the domain size (*i.e.,* the number of actions). To achieve efficient exploration, we propose to collect part of the training trajectories through a helper demo

policy. In practice, one can resort to a policy induced by any off-the-shelf rule-based AFs, such as Expected Hypervolume Improvement (Emmerich & Klinkenberg, 2008).

Notably, as a general-purpose multi-objective optimization solver, BOFormer enjoys the following salient features: (i) *No hypervolume identifiability issue*: Built on the proposed Generalized DQN framework, BOFormer systematically addresses the identifiability issue such that it can approximately recover the Pareto front under a small sampling budget. (ii) *Zero-shot transfer*: BOFormer is trained solely on synthetic GP functions and can achieve zero-shot transfer to other unseen testing functions. With that said, it does not require any fine-tuning or any metadata during inference at deployment. (iii) *Cross-domain transfer capability*: BOFormer nicely supports cross-domain transfer in the sense that the domain size and dimensionality of the black-box functions for training can be different from those of the testing functions. (iv) *No Monte-Carlo estimation needed during inference*: As a learning-based AF, BOFormer completely obviates the need for the computationally heavy Monte-Carlo estimation required by many rule-based AFs and thereby enjoys efficient inference during deployment. The main contributions could be summarized as follows:

- We identify the critical hypervolume identifiability issue due to the inherent non-Markovianity in learning the AFs for MOBO. To resolve this, inspired by the literature of general RL, we present the Generalized DQN framework for non-Markovian environments.
- To substantiate the Generalized DQN framework, we propose BOFormer, which leverages the Transformer architecture and reinterprets MOBO as a sequence modeling problem. To the best of our knowledge, BOFormer serves as the first RL-based AF for MOBO. Moreover, several practical enhancements are proposed to facilitate the training of BOFormer.
- We construct a hypervolume optimization dataset called HPO-3DGS, consisting of 68000 different parameters on 3D Gaussian Splatting (Kerbl et al., 2023) for 3D object reconstruction problems with 5 different scenes. We evaluate the proposed BOFormer on a variety of black-box functions, including both synthetic optimization functions and real-world hyperparameter optimization on HPO-3DGS. We demonstrate that the proposed AF significantly outperforms both existing rule-based AFs and other Transformer-based RL benchmark methods.

## 2 RELATED WORK

### 2.1 MULTI-OBJECTIVE BAYESIAN OPTIMIZATION

**Random Scalarization.** To leverage the plethora of AFs for SOBO in the MOBO setting, random scalarization addresses MOBO via iteratively solving single-objective BO subproblems under a scalarization function, such as a direct weighted sum or the Tchebycheff scalarization function (Miettinen, 1999). Notably, random scalarization was originally developed for recovering the Pareto front under evolutionary methods, such as the celebrated ParEGO (Knowles, 2006), MOEA/D (Zhang & Li, 2007), and RVEA (Cheng et al., 2016), and has been subsequently adapted to solving MOBO (Zhang et al., 2009; Paria et al., 2020). Despite its simplicity, as random scalarization enforces exploration mainly by the random sampling of the scalarization parameters, this approach is known to be sensitive to the scale of the different objective functions and could suffer under high-dimensional search spaces (Daulton et al., 2022).

**Improvement Maximization.** Another popular class of AFs is built on the maximization of improvement-based metrics, such as the expected one-step improvement in hypervolume (EHVI) in (Emmerich & Klinkenberg, 2008; Emmerich et al., 2011; Hupkens et al., 2015; Yang et al., 2019) (also known as the $\mathcal{S}$-metric in (Beume et al., 2007; Ponweiser et al., 2008)), sequential uncertainty reduction (Picheny, 2015), and the hypervolume knowledge gradient (Daulton et al., 2023). However, evaluating the one-step expected improvement typically involves a multi-dimensional integral, which is difficult to derive directly and hence needs to be approximated by the costly Monte Carlo estimation. Accordingly, to tackle the above computational complexity issue, differentiable methods have recently been developed to enable fast parallel evaluations of these AFs, such as $q$EHVI (Wada & Hino, 2019; Daulton et al., 2020) and $q$NEHVI (Daulton et al., 2021) in BoTorch (Balandat et al., 2020).

**Information-Theoretic Search Methods.** Various information-theoretic criteria have been utilized in the context of MOBO. For example, Hernández-Lobato et al. (2016) proposes Predictive Entropy Search for MOBO (PESMO), which selects the candidate point with maximal reduction in the entropy of the posterior distribution over the Pareto-optimal input set, and is subsequently extended to the constrained setting (Garrido-Merchán & Hernández-Lobato, 2019). Subsequently, Belakaria et al.

(2019) propose MESMO, which extends the Max-value Entropy Search approach (Wang & Jegelka, 2017) to the principle of output space entropy search for MOBO, *i.e.,* utilizes the information gain about the Pareto-optimal output set as a more computationally tractable sampling criterion (Belakaria et al., 2021). Suzuki et al. (2020) propose Pareto-Frontier Entropy Search, which utilizes information gain of the Pareto front in the AF design. Moreover, Joint Entropy Search (JES) further takes into account the joint information gain of the Pareto-optimal set of inputs and outputs (Tu et al., 2022; Hvarfner et al., 2022). On the other hand, USeMO (Belakaria et al., 2020) uses the volume of the uncertainty hyper-rectangle as an alternative uncertainty measure for sampling.

Despite the plethora of AFs developed for MOBO, most of them are built on optimizing one-step information-theoretic metrics and do not explore the possibility of multi-step look-ahead policies. By contrast, the proposed BOFormer takes the overall long-term effect of each sample into account through non-Markovian RL and sequence modeling.

## 2.2 SINGLE-OBJECTIVE BLACK-BOX OPTIMIZATION VIA LEARNING

Several recent attempts have tackled SOBO problems from the perspective of RL-based AFs. Volpp et al. (2020) propose MetaBO, which leverages actor-critic RL to learn AFs from GP functions for transfer learning. Subsequently, Hsieh et al. (2021) proposes a meta-RL framework termed Few-Shot Acquisition Function (FSAF), which learns a Bayesian deep Q-network as a differentiable AF and adapts the Bayesian model-agnostic meta-learning (Yoon et al., 2018) in order to enable few-shot fast adaptation to various black-box functions based on metadata. (Shmakov et al., 2023) proposes to solve SOBO through a combination of transformer-based deep kernels and RL-based acquisition functions. Despite the above, the existing solutions all focus on SOBO under the standard RL formulation and therefore cannot be directly applied to the non-Markovian problem of MOBO. Moreover, as optimization of single-objective black-box functions is essentially a sequential decision making problem, several recent attempts manage to learn sequence models in an end-to-end manner. For example, Chen et al. (2022) propose OptFormer, which focuses on hyperparameter optimization (HPO) and leverages Transformers through fine-tuning on an offline dataset to enable adaptation to the HPO tasks. More recently, Maraval et al. (2023) propose Neural Acquisition Process (NAP), which is a multi-task variant of Neural Process (NP) simultaneously learning an acquisition function and the predictive distributions, without using the surrogate GP model. To the best of our knowledge, our paper offers the first learning-based solution to MOBO.

**Remarks on Application Scope and Objectives.** Notably, there are two salient differences between BOFormer and the above two works: (i) *Application scope*: BOFormer is positioned as a general-purpose multi-objective black-box optimization solver with superior cross-domain capability (*i.e.,* the size and the dimensionality of the input domains can be different between the training and deployment phases). In contrast, OptFormer is designed specifically for HPO, and NAP is built on the idea of transfer learning in BO and has limited cross-domain transferability. (ii) *Multiple objectives*: Both OptFormer and NAP focus on single-objective problems and are not readily applicable to recovering the Pareto front in the multi-objective setting. By contrast, BOFormer directly tackles multi-objective optimization and addresses the inherent identifiability issue. Therefore, we consider the above OptFormer and NAP as orthogonal directions to ours. Moreover, the proposed BOFormer can also benefit from the learned neural process in NAP and other variants of NPs (Garnelo et al., 2018; Kim et al., 2018) as surrogate models beyond GPs.

Due to the page limit, we defer the related works on sequence modeling for RL to Appendix B.1.

## 3 PRELIMINARIES

In this section, we present the formulation of MOBO and the background of general RL. Throughout this paper, we let $\Delta(\mathcal{Z})$ denote the set of all probability distributions over a set $\mathcal{Z}$ and use $[K]$ as a shorthand for $\{1, \cdots, K\}$.

### 3.1 MULTI-OBJECTIVE BAYESIAN OPTIMIZATION

The goal of MOBO is to design an algorithm that sequentially takes samples from the input domain $\mathbb{X} \subset \mathbb{R}^d$ to jointly optimize a black-box vector-valued function $\boldsymbol{f} : \mathbb{X} \to \mathbb{R}^K$, under a sampling budget $T \in \mathbb{N}$. For ease of exposition, we also write $\boldsymbol{f}(x) := (f_1(x), \cdots, f_K(x))$ as the tuple of the $K$

scalar objective functions, for each $x \in \mathbb{X}$. At each step $t$, the algorithm selects a sample point $x_t \in \mathbb{X}$ and observes the corresponding function values $\boldsymbol{y}_t := (y_t^{(1)}, \cdots, y_t^{(K)})$, where $y_t^{(i)} = f_i(x_t) + \varepsilon_{t,i}$ is the noisy observation of the $i$-th entry of the function output and $\varepsilon_{t,i}$'s are i.i.d. zero-mean Gaussian noises. For notational convenience, we use $\mathcal{F}_t := \{(x_i, \boldsymbol{y}_i)\}_{i \in [t]}$ to denote the observations up to $t$.

**Pareto Front and Hypervolume.** To construct a (partial) ordering over the points of the input domain, we say that $\boldsymbol{f}(x)$ *dominates* $\boldsymbol{f}(x')$ if $f_i(x) \geq f_i(x')$ for all $i \in [K]$ and $f_j(x) > f_j(x')$ for at least one element $j$. For simplicity, we write $x \succ x'$ if $\boldsymbol{f}(x)$ dominates $\boldsymbol{f}(x')$. Based on this, the *Pareto front* (denoted by $\mathcal{X}^*$) is defined as the subset of $\mathbb{X}$ that cannot be dominated by any other point in $\mathbb{X}$, *i.e.,* $\mathcal{X}^* := \{x \in \mathbb{X} | x' \not\succ x, \forall x' \in \mathbb{X}\}$. An alternative description of the goal of MOBO is to discover the Pareto front. Accordingly, MOBO algorithms are typically evaluated from the perspective of *hypervolume*, which offers a natural performance metric for capturing the inherent trade-off among different objective functions. Specifically, given a reference point $\boldsymbol{u} \in \mathbb{R}^K$ and any subset $\mathcal{X} \subseteq \mathbb{X}$, the hypervolume of $\mathcal{X}$ is defined as (Zitzler & Thiele, 1999):

$$\text{HV}(\mathcal{X}; \boldsymbol{u}) := \lambda\bigg( \bigcup_{x \in \mathcal{X}} \big\{ \boldsymbol{y} | \boldsymbol{f}(x) \succ \boldsymbol{y} \succ \boldsymbol{u} \big\} \bigg),$$

where $\lambda(\cdot)$ is the $K$-dimensional Lebesgue measure and $\boldsymbol{y}$s are those elements that satisfy $\boldsymbol{f}(x) \succ \boldsymbol{y} \succ \boldsymbol{u}, x \in \mathcal{X}$. In practice, the reference point can be configured as $\boldsymbol{u} = (\min_{x \in \mathbb{X}} f_1(x), \cdots, \min_{x \in \mathbb{X}} f_k(x))$. To evaluate a policy, we consider the *simple regret* defined as $\mathcal{R}(t) := \text{HV}(\mathbb{X}) - \text{HV}(\mathcal{X}_t)$, which measures the overall performance of the samples up to time step $t$. For brevity, we simply use $\text{HV}(\mathcal{X})$ as a shorthand for $\text{HV}(\mathcal{X}; \boldsymbol{u})$ in the sequel.

**Gaussian Process as a Surrogate Model.** To maximize hypervolume in a sample-efficient manner, MOBO imposes a function prior through GP, which serves as a surrogate probabilistic model for capturing the underlying structure of the objective functions. Specifically, as a Bayesian approach, the GP assumes that for each objective function $f_i(\cdot)$, the function values at any set of input points form a multivariate Gaussian distribution, which can be fully characterized by a mean function and a covariance kernel. Therefore, under a GP prior, given the observations $\mathcal{F}_t$ up to time $t$, the posterior predictive distribution of each $f_i(x)$ ($x \in \mathbb{X}$) remains Gaussian and can be written as $\mathcal{N}(\mu_t^{(i)}(x), \sigma_t^{(i)}(x)^2)$, where $\mu_t^{(i)}(x) := \mathbb{E}[f_i(x) | \mathcal{F}_t]$ and $\sigma_t^{(i)}(x) := \sqrt{\mathbb{V}[f_i(x) | \mathcal{F}_t]}$ can be derived in closed form through matrix operations (Williams & Rasmussen, 2006). For notational convenience, we let $\boldsymbol{\mu}_t(x) := (\mu_t^{(1)}(x), \cdots, \mu_t^{(K)}(x))$ and $\boldsymbol{\sigma}_t(x) := (\sigma_t^{(1)}(x), \cdots, \sigma_t^{(K)}(x))$.

**Acquisition Functions.** With the help of GPs, one natural way to address BO is through planning as in optimal control (*e.g.,* via dynamic programming). However, finding the exact optimal policy for BO (either single- or multi-objective) is known to be computationally intractable in general due to the curse of dimensionality. To tackle this issue, BO resorts to *index-type* strategies induced by an acquisition function $\Upsilon(\boldsymbol{\mu}_t(x), \boldsymbol{\sigma}_t(x))$, which takes the posterior mean and variance as input and outputs an indicator for quantifying the usefulness of each potential candidate sample point $x \in \mathbb{X}$, typically based on some handcrafted design criteria. For example, the celebrated Expected Hypervolume Improvement (EHVI) method constructs an AF as $\Upsilon_{\text{EHVI}}(\boldsymbol{\mu}_t(x), \boldsymbol{\sigma}_t(x)) := \mathbb{E}[\text{HV}(\mathcal{X}_t \cup \{x\}) - \text{HV}(\mathcal{X}_t) | \mathcal{F}_t]$, which involves a multi-dimensional integral with respect to the posterior distribution characterized by $\boldsymbol{\mu}_t(x)$ and $\boldsymbol{\sigma}_t(x)$.

### 3.2 GENERAL RL IN NON-MARKOVIAN ENVIRONMENTS

To achieve RL without Markovianity, several generalizations of the standard Markov decision process (MDP) have been proposed, such as the classic partially-observable MDPs (Monahan, 1982; Jaakkola et al., 1994), the early works on general RL (Lattimore et al., 2013; Leike, 2016; Majeed, 2021), and the more recent attempts on RL for arbitrary environments (Dong et al., 2022; Lu et al., 2023; Bowling et al., 2023). In this paper, we consider the general RL formulation in (Dong et al., 2022; Lu et al., 2023) to address policy learning beyond Markovianity.

**Environment.** The general interaction protocol of the agent and the environment can be described as follows. Let $\mathcal{A}$ and $\mathcal{O}$ denote the set of actions and observations, respectively. At each time $t \in \mathbb{N}$, the agent first receives a new observation $O_t \in \mathcal{O}$ from the environment and takes an action $A_t \in \mathcal{A}$ based on the history $H_t := (A_0, O_1, A_1, \cdots, A_{t-1}, O_t)$ observed so far. For simplicity, we let the initial history $H_0$ be empty. We also define the set of all $n$-step histories as $\mathcal{H}^{(n)} := (\mathcal{A} \times \mathcal{O})^n$

and accordingly define the set of all finite histories as $\mathcal{H} := \bigcup_{n \geq 0} \mathcal{H}^{(n)}$. The transition dynamics of the environment is captured by the *transition function* $p : \mathcal{H} \times \mathcal{A} \rightarrow \Delta(\mathcal{O})$, which determines the transition probability $p(o|h, a) \equiv \mathbb{P}(O_{t+1} = o|H_t = h, A_t = a)$ of observing $o$ upon applying action $a$ under history $h$. Moreover, let $r : \mathcal{H} \times \mathcal{A} \times \mathcal{O} \rightarrow [-r_{\max}, r_{\max}]$ denote the reward function. Notably, the reward function $r$ in non-Markovian environments is allowed to be history-dependent and hence better suits the MOBO problems. Let $\gamma \in [0, 1)$ denote the discount factor for the rewards.

**Policies and Value Functions.** The agent specifies its strategy through a *policy* $\pi : \mathcal{H} \rightarrow \Delta(\mathcal{A})$, which maps each history to a probability distribution over the action set. Let $\Pi$ denote the set of all policies. Similar to the MDP setting, we define value functions that reflect the long-term benefit of following a policy $\pi$. Given a $\tau$-step history $h \in \mathcal{H}$,

$$V^\pi(h) := \mathbb{E}_\pi\Big[\sum_{t=\tau}^\infty \gamma^{t-\tau} r(H_t, A_t, O_{t+1})\Big|H_\tau = h\Big],$$

$$Q^\pi(h, a) := \mathbb{E}_\pi\Big[\sum_{t=\tau}^\infty \gamma^{t-\tau} r(H_t, A_t, O_{t+1})\Big|H_\tau = h, A_\tau = a\Big].$$

Moreover, we extend the definitions of the optimal value functions in MDPs to the non-Markovian setting as

$$V^*(h) := \sup_{\pi \in \Pi} V^\pi(h), \; Q^*(h, a) := \sup_{\pi \in \Pi} Q^\pi(h, a). \tag{1}$$

The proposition below offers a generalized version of the Bellman optimality equations and characterizes $V^*$ and $Q^*$.

**Proposition 3.1** (Dong et al. (2022)). *The pair of $(V^*, Q^*)$ is the unique solution to the following system of equations:*

$$V(h) = \max_{a' \in \mathcal{A}} Q(h, a') \tag{2}$$

$$Q(h, a) = \mathbb{E}_{o \sim p(\cdot|h,a), h' \equiv (h,a,o)}\big[r(h, a, o) + \gamma V(h')\big], \tag{3}$$

*where $V : \mathcal{H} \rightarrow \mathbb{R}$ and $Q : \mathcal{H} \times \mathcal{A} \rightarrow \mathbb{R}$ are bounded real-valued functions.*

## 4 METHODOLOGY

### 4.1 GENERALIZED DQN FOR NON-MARKOVIAN PROBLEMS

Motivated by the optimality equations in (2)-(3), we convert these fundamental properties into a learning algorithm.

**Loss Function.** To learn $Q^*$, we adapt the loss function of the standard DQN to the generalized non-Markovian version by minimizing the residual of the optimality equation. Let $Q_\theta(h, a)$ denote the parameterized Q-function. Then, the loss function of the generalized DQN is designed as

$$\mathbb{E}_{(h,a,o) \sim \mathcal{D}}\Big[\Big(r(h, a, o) + \gamma \max_{a' \in \mathcal{A}} Q_{\bar{\theta}}(h', a') - Q_\theta(h, a)\Big)^2\Big], \tag{4}$$

where $\mathcal{D}$ is the underlying distribution of the observed histories during training, $h' = (h, a, o)$ is the history for the next Q-value, and $Q_{\bar{\theta}}$ is a copy of $Q_\theta$ with parameters frozen.

**Remark 4.1.** The above loss function bears some resemblance to that of the POMDP variant of DQN, such as Deep Recurrent Q-Networks (DRQN) in (Hausknecht & Stone, 2015). Despite this, one fundamental difference is: The POMDP formulation presumes that there exists a hidden true state, which determines the transitions and the reward function, and the hidden state is to be learned and deciphered by recurrent neural networks in DRQN. By contrast, Generalized DQN does not make this presumption and involves only the history of observations and actions.

**Direct Implementation of Generalized DQN.** To implement (4), one natural design is to leverage sequence modeling (*e.g.,* Transformers) and directly use the full observations as the input of the sequence models. This design principle is widely adopted in Transformer-based RL (Chen et al., 2021; Janner et al., 2021; Chebotar et al., 2023) for various popular RL benchmark tasks (*e.g.,* locomotion

and robot arm manipulation in MuJoCo (Todorov et al., 2012)). In the context of learning AFs for MOBO, one can apply this design principle and extend the representation design of AF for SOBO (cf. Figure 1) to the MOBO setting, and this amounts to taking the posterior distributions of all $K$ objective functions at all the domain points along with $\{y_t^{(i)*} := \text{argmax}_{j \leq t-1} y_j^{(i)}\}_{i=i}^K$ the best function values observed so far as the per-step observation, *i.e.,* $o \equiv \{\mu^{(i)}(x), \sigma^{(i)}(x), y^{(i)*}\}_{x \in \mathcal{X}, i \in [K]}$. While being a natural variant of Transformer-based RL, this implementation of the Generalized DQN framework can be problematic in MOBO for two reasons: (i) *Limited cross-domain transferability*: As the observation representation is domain-dependent under this design, the learned model is tied closely to the training domain and has very limited transferability. As a result, retraining or customization is needed for every task at deployment. (ii) *Scalability issue in sequence length and memory requirement:* Under this design, the sequence length would grow linearly with the number of domain points and pose a stringent requirement on the hardware memory for training. Indeed, the domain size is at least on the order of thousands in practical BO problems (*e.g.,* circuit design (Lyu et al., 2018) and hyperparameter optimization (Lindauer et al., 2022)).

To tackle the above issues, we propose an alternative design that better substantiates the Generalized DQN framework for MOBO with domain-agnostic representations and several practical enhancements, as detailed in Section 4.2.

## 4.2 BOFORMER: AN ENHANCED IMPLEMENTATION OF GENERALIZED DQN

To avoid the issues of the direct implementation, we propose BOFormer, which is built on the following enhancements. The pseudo code is in Algorithm 1 in Appendix C.

*Q*-**Augmented Representation.** Define

$$y_t^{(i)*} := \max_{1 \leq j \leq t} y_i^{(i)}, \forall i \in [1, \cdots, K]$$

as the best observed function value of $i$-th objective at time $t$. Moreover, for each domain point $x \in \mathbb{X}$, let $o_t(x)$ denote the observation for $x$ as

$$o_t(x) \equiv \left\{\mu_t^{(i)}(x), \sigma_t^{(i)}(x), y_t^{(i)*}, \frac{t}{T}\right\}_{i \in [K]}$$

Moreover, in BOFormer, we use the normalized hypervolume improvement as the reward, *i.e.,*

$$r_t := \frac{\text{HV}(\mathcal{X}_t) - \text{HV}(\mathcal{X}_{t-1})}{\text{HV}(\mathcal{X}^*) - \text{HV}(\mathcal{X}_t)}.$$

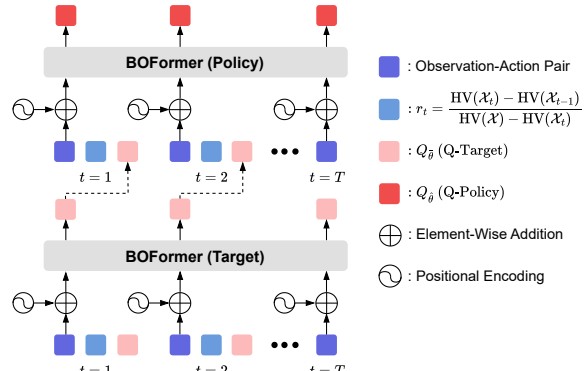

Figure 2: BOFormer comprises two distinct networks as shown above: The upper network functions as the policy network, utilizing the historical data and the Q-value predicted by the target network to estimate the Q-values for action selection. The lower network serves as the target network, responsible for constructing Q-values for past observation-action pairs.

Then, $h_t$, the history up to time $t$, is the concatenation of past observation-action pair representation defined as follows:

$$h_t = \left\{\mu_j^{(i)}(x_j), \sigma_j^{(i)}(x_j), y_{j-1}^{(i)*}, j/t, r_i, Q_{\bar{\theta}}\right\}_{i \in [k], j \in [t-1]}. \tag{5}$$

Notably, under this design, the representation is domain-agnostic and memory-efficient in the sense that its dimension does not increase with the domain size.

**BOFormer as an Acquisition Function for MOBO.** The model structure of BOFormer is provided in Figure 2. Denote $Q_\theta(\cdot) : \mathcal{H}^{(t-1)} \times \mathcal{O} \to \mathbb{R}$ to be the function of BOFormer parameterized by $\theta$ and let $\widehat{\theta}$ represent the parameters of BOFormer. The selected point $x_t$ satisfies that $x_t := \text{argmax}_{x \in \mathbb{X}} Q_{\widehat{\theta}}(h_t, o_t(x))$.

Then, $Q_{\bar{\theta}}$ considered in $h_t$ can be implemented by a target network, as is commonly done in Deep Q-learning. In non-Markovian version, $Q_{\bar{\theta}}$ can be defined recursively, where

$$Q_{\bar{\theta}}(h_t, o_t(x_t)) := Q_{\bar{\theta}}\left(\{o_j(x_j), r_j, Q_{\bar{\theta}}(h_{j-1}, o_{j-1}(x_{j-1}))\}_{j=1}^{t-1}, o_t(x_t)\right).$$

**Remark 4.2.** To handle continuous domains, BOFormer can approximate the maximum Q-value using Sobol grids, as adopted by (Volpp et al., 2020; Hsieh et al., 2021). However, this could be a limitation in high-dimensional tasks. To address this, an empirical study on high-dimensional problems and the detailed procedure of employing Sobol grids are in Appendix D.3 and C.2, respectively.

**Off-Policy Learning and Prioritized Trajectory Replay Buffer.** We extend the concept of Prioritized Experience Replay (PER) (Schaul et al., 2016) and introduce the Prioritized Trajectory Replay Buffer (PTRB). The detailed modifications are as follows: (i) Elements pushed into this buffer are entire trajectories $\tau = \{o_i(x_i), r_i\}_{i=1}^T$. (ii) The TD-error considered in PER is replaced by $\delta(Q_{\theta_t}, \tau)$, which is the summation of the TD-error of the policy network for all transitions in this trajectory, *i.e.,*

$$\delta(Q, \tau) := \sum_{i=1}^{T-1} \left( Q\left(h_i, o_i(x_i)\right) - \left(r_i + \gamma \max_{x \in \mathbb{X}} Q_{\bar{\theta}}(h_{i+1}, o_{i+1}(x))\right)\right)^2. \tag{6}$$

Let $\mathcal{B}$ denote the batch sampled from PTRB. The loss function of BOFormer is defined as $L(\theta) := \sum_{\tau \in \mathcal{B}} \delta(Q_\theta, \tau)$.

## 5 EXPERIMENTS

In this section, we evaluate the proposed BOFormer against popular MOBO methods on both synthetic and hyperparameter optimization on 3D Gaussian Splatting (3DGS) (Kerbl et al., 2023). Unless stated otherwise, we report the average attained hypervolume at the final step over 100 evaluation episodes in the main text. Due to the space limit, all the statistics, including the hypervolume at each step and performance profiles (Agarwal et al., 2021) are provided in the Appendix.

**Benchmark Methods.** We compare BOFormer with various classic and state-of-the-art benchmark methods, including: (i) *Rule-based methods*: NEHVI (Daulton et al., 2021), ParEGO (Knowles, 2006), NSGA-II (Deb et al., 2002), HVKG (Daulton et al., 2023), and JES (Tu et al., 2022; Hvarfner et al., 2022). Regarding NEHVI, ParEGO, and HVKG, we use the differentiable Monte-Carlo version, namely $q$NEHVI, $q$ParEGO, and $q$HVKG, provided by BoTorch (Balandat et al., 2020). (ii) *Learning-based methods*: Given that BOFormer is the first learning-based MOBO method, we consider the direct multi-objective extension of FSAF (Hsieh et al., 2021), which achieves state-of-the-art results in SOBO. To showcase the design of BOFormer, we also adapt a popular RL Transformers, namely Decision Transformer (DT) (Chen et al., 2021), to the MOBO setting. Moreover, we also include Q-Transformer (QT) (Chebotar et al., 2023), a more recent Transformer design RL that uses a similar DQN loss (termed Autoregressive Discrete Q-Learning in their paper) and can be viewed as a variant of BOFormer without Q-augmented representation. To further demonstrate the competitiveness of BOFormer, we also compare it with OptFormer (Chen et al., 2022), a recent Transformer-based method designed specifically for hyperparameter optimization. All the learning-based methods are trained on GP functions under with the lengthscales drawn randomly from $[0.1, 0.4]$ for fairness. The detailed configuration is provided in Appendix A.

*Q: Does BOFormer achieve sample-efficient MOBO on a variety of optimization problems?*

**Synthetic Functions.** We answer this question by first evaluating BOFormer extensively on a diverse collection of synthetic black-box functions: (i) Combinations of functions with many local optima, including *Ackley-Rastrigin (ARa)*; (ii) Combination of smooth functions, including *Branin-Currin (BC) and Branin-Currin-Dixon*; (iii) Combination of non-smooth and smooth functions, including *Ackley-Rosenbrock (AR) and Dixon-Rastrigin (DRa)* Table 1 and Figure 6 shows the averaged hypervolume on synthetic problems. Based on Figure 6, we can observe that BOFormer constantly achieves the largest or among the largest hypervolume among all methods. In the cases of Branin and Currin, the performance of BOFormer is not as expected, which we attribute to the larger length scales of these functions compared to others. Then, QT, a variant of the generalized DQN but without Q-augmented representation, does not perform well. This indicates that sequence modeling itself does not necessarily guarantee an efficient search for the Pareto front, and Q-augmented representation are needed for solving MOBO, as showcased by the proposed BOFormer. Figure 3 shows the performance profiles, which are meant to more reliably present the performance variability of BOFormer and other baselines across testing episodes than the interval estimates of aggregate metrics. We observe that in most tasks, the profiles of BOFormer sit on the top right of the other baselines and hence enjoy a statistically better performance in hypervolume. Please see Appendix D for more performance profiles of the final attained hypervolume.

**Multi-Objective Hyperparameter Optimization on 3D Gaussian Splatting.** We also evaluate our method on 3D object reconstruction problems. This task takes multiple images, which captures an object, as input, and the goal is to reconstruct a 3D representation for novel-view rendering. 3D Gaussian Splatting (3DGS) by Kerbl et al. (2023) is the recent state-of-the-art on this task. One practical issue of 3DGS is their sensitivity to the hyperparameters. To achieve the best quality, we typically require manual tuning for each capture, which is time consuming and requires expert knowledge. An automatically hyperparameters tuning pipeline thus become very useful. To evaluate on this task, we construct a dataset by consulting domain expert and dense-grid searching 1440 hyperparameters of 3DGS. We perform 30 samples for this task where we defer the details in the supplementary. The objectives are negative model size and novel-view rendering quality measured by PSNR. We use 4 different objects from (Mildenhall et al., 2021) and 64 different chairs from (Yu et al., 2023) to compare the performance of different hyperparameter tuning methods. Again, from Table 2 and Figure 7, we observe that BOFormer remains the best or among the best in all the tasks. This demonstrates the wide applicability of the proposed BOFormer. The detailed hypervolume per step and performance profile of the final hypervolume for 3DGS dataset is provided in Appendix D.

Table 1: The average attained hypervolume at the 100th sampling step under synthetic functions. Boldface and underlining denote performance within 1% of the best-performing method.

| | AR | ARa | BC | DRa | RBF | Matern | BCD |
|---|---|---|---|---|---|---|---|
| *Rule-based Methods* | | | | | | | |
| qHVKG (Daulton et al., 2023) | 0.5787 | 0.7008 | **0.4751** | **0.9499** | 0.8646 | 0.8506 | 0.3547 |
| NSGA-II (Deb et al., 2002) | 0.5557 | 0.7122 | 0.4271 | **0.9573** | 0.8603 | 0.8569 | 0.3400 |
| qNEHVI (Daulton et al., 2021) | 0.5428 | 0.6290 | **0.4773** | 0.9333 | **0.8731** | **0.8696** | **0.3697** |
| JES (Tu et al., 2022) | 0.4930 | 0.6207 | 0.4487 | 0.9392 | 0.8661 | 0.8594 | 0.3345 |
| qParEGO (Knowles, 2006) | 0.5410 | 0.6111 | 0.4597 | 0.9347 | **0.8730** | **0.8684** | 0.3564 |
| *Learning-based Methods* | | | | | | | |
| BOFormer (Ours) | **0.5900** | **0.7377** | 0.4476 | 0.9461 | **0.8751** | **0.8642** | 0.3617 |
| FSAF (Hsieh et al., 2021) | 0.4424 | 0.4800 | 0.4175 | 0.9193 | 0.8381 | 0.8566 | 0.3578 |
| OptFormer (Chen et al., 2022) | 0.4448 | 0.4835 | 0.4143 | 0.8927 | 0.8558 | 0.8488 | 0.3437 |
| DT (Chen et al., 2021) | 0.4397 | 0.4855 | 0.4159 | 0.9141 | 0.8409 | 0.8359 | 0.2980 |
| QT (Chebotar et al., 2023) | 0.4630 | 0.4953 | 0.4086 | 0.8584 | 0.8685 | 0.8550 | 0.3320 |

Table 2: The average hypervolume at the 30th step under 3DGS hyper-parameter optimization scenarios. Boldface and underlining denote performance within 1% of the best-performing method.

| | Chairs | Lego | Materials | Mic | Ship |
|---|---|---|---|---|---|
| *Rule-based Methods* | | | | | |
| qHVKG (Daulton et al., 2023) | 0.8508 | 0.9365 | 0.9088 | 0.8234 | 0.9630 |
| NSGA-II (Deb et al., 2002) | 0.8500 | 0.9192 | 0.8831 | 0.8051 | 0.9615 |
| qNEHVI (Daulton et al., 2021) | **0.9159** | 0.9344 | 0.9041 | 0.8353 | **0.9661** |
| JES (Tu et al., 2022) | 0.9003 | **0.9409** | 0.9217 | 0.8251 | **0.9656** |
| qParEGO (Knowles, 2006) | **0.9103** | 0.9335 | 0.9098 | 0.8263 | 0.9635 |
| *Learning-based Methods* | | | | | |
| BOFormer (Ours) | **0.9162** | **0.9508** | **0.9224** | **0.8816** | **0.9745** |
| FSAF (Hsieh et al., 2021) | **0.9120** | **0.9504** | **0.9213** | **0.8871** | **0.9737** |
| OptFormer (Chen et al., 2022) | 0.8200 | 0.7875 | 0.7838 | 0.7865 | 0.9575 |
| DT (Chen et al., 2021) | 0.8602 | 0.9315 | 0.9005 | **0.8786** | 0.9409 |
| QT (Chebotar et al., 2023) | 0.9119 | 0.9368 | 0.9142 | 0.8731 | 0.9551 |

***Q: The comparison between BOFormer and OptFormer.*** From Tables 1-2 and Figures 3-8, we observe that the performance of BOFormer surpasses that of OptFormer. We conjecture that the reasons are two-fold: (i) OptFormer takes a supervised learning perspective to learn the context that describes the HPO information while BOFormer leverages non-Markovian RL for better long-term planning. (ii) OptFormer reinterprets HPO as a language modeling problem in a text-to-text manner. This approach necessitates that the training dataset closely resembles the testing domain, and this requirement does not hold here (*e.g.,* dimensionality of the training domain differs from that of the testing domain).

***Q: A study on the effect of sequence length of BOFormer.*** We also conducted an ablation study on evaluating how the sequence length (denoted by $w$) would affect the hypervolume performance of BOFormer. Figure 4 shows that the hypervolume of non-Markovian BOFormer ($w > 1$) is superior to that of Markovian BOFormer ($w = 1$). The results also demonstrate that sequential modeling can successfully solve the hypervolume identifiability issue.

***Q: A study on computational efficiency.*** We provide computation times per step in Table 3. BOFormer and qNEHVI are competitive in terms of final hypervolume (Tables 1-2), with the computation times

of BOFormer being less sensitive to the number of objective functions and shorter than those of qNEHVI under the 3-objective function setting.

***Q: How is the transfer ability of BOFormer under different numbers of objective functions?***
Although learning-based methods generally require training multiple models for different numbers of objective functions due to mismatched model structures, we conducted an experiment on BOFormer to determine whether it is possible to train only the state-action embedding layer while utilizing a pre-trained transformer model from a different number of objective functions. The results, shown in Figure 8, demonstrate that BOFormer exhibits excellent transfer ability across different numbers of objective functions. The detailed implementation of transferring on BOFormer is in Appendix C.2.

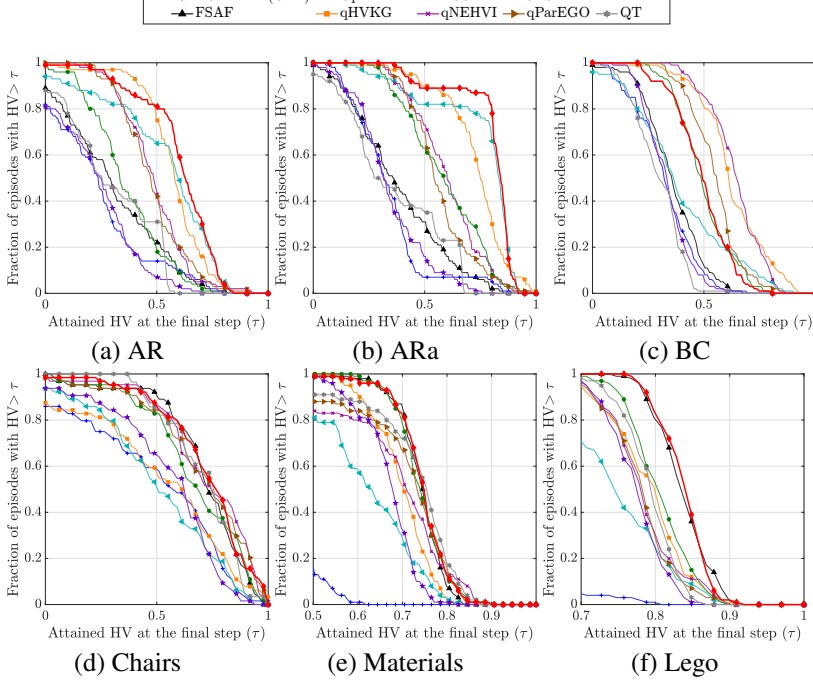

Figure 3: Performance profiles of hypervolume at the final step.

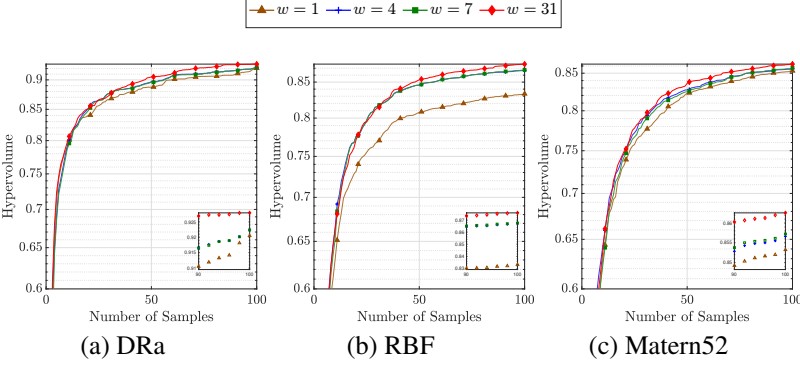

Figure 4: Attained hypervolumes of BOFormer under various sequence lengths.

# 6 CONCLUSION

In this paper, we address MOBO problems from the perspective of RL-based AF by identifying and tackling the inherent hypervolume identifiability issue. We achieve this goal by first presenting a generalized DQN framework and implementing it through BOFormer, which leverages the sequence modeling capability of Transformers and incorporates multiple enhancements for MOBO. Our experimental results show that BOFormer is indeed a promising approach for general-purpose multi-objective black-box optimization.

ACKNOWLEDGMENTS

This material is based upon work partially supported by NVIDIA grant and technical involvement from NVIDIA Taiwan AI R&D Center (TRDC), National Science and Technology Council (NSTC), Taiwan under Contract No. NSTC 113-2628-E-A49-026, and the Higher Education Sprout Project of the National Yang Ming Chiao Tung University and Ministry of Education (MOE), Taiwan. We also thank the National Center for High-performance Computing (NCHC) for providing computational and storage resources.

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

APPENDICES

# Table of Contents

## A  DETAILED EXPERIMENTAL CONFIGURATION

### A.1  ENVIRONMENT

In each testing episode, the agent interacts with the environment for a total of $T = 100$ for synthetic functions and $T = 30$ time steps for HPO-3DGS. The observed function values are subject to noise $\epsilon \sim N(0, 0.1)$ for synthetic functions, while the function values for HPO-3DGS are observed without noise. During testing and training, the objective functions of each episode are scaled by perturbation noise $\kappa$, which is sampled from $\text{Uniform}(0, 0.1)$ for synthetic functions and $\text{Uniform}(0, 0.01)$ for other functions.

For HPO-3DGS, there are about 68,000 data points related to 4 different objects: Lego, Materials, Mic, Ship, and Chairs. Additionally, there are 64 different scenes involving chairs. While testing on chairs, each episode is conducted with an individual scene.

**Configuration of the Surrogate Model:**

- Training: For all the learning-based algorithms, we used the same GP surrogate model with an RBF kernel for obtaining the posterior distributions. Regarding the generation of GP synthetic functions as objective functions during training, each function is generated randomly under either an RBF or Matern-5/2 kernel with lengthscale sampled from $[0.1, 0.4]$.

- Testing: During testing, all algorithms utilized a surrogate model based on the Matern52 kernel. The length scale was estimated online by maximizing the marginal likelihood by following the same approach as BoTorch.

### A.2 Hyperparameters of Learning-Based Approaches

- **BOFormer:** hidden size: 128 for all linear layers used to embed positional encodings, state-action pairs, rewards, and Q-values, learning rate: $10^{-5}$, weight decay: $10^{-5}$, $r_{\text{demo}}$: 0.01, batch size: 8, number of attention layer: 8, number of head of attention layer: 4, window size $w = 31$, dropout: 0.1, buffer size: 64, training episode: 3000.

- **DT**[*]**:** hidden size: 128, learning rate: $10^{-4}$, weight decay: $10^{-4}$, batch size: 16, number of attention layer: 3, number of head of attention layer: 2, embed dim: 500, dropout: 0.1, warmup steps: 10, max length: 100.

- **QT:** hidden size: 128, learning rate: $10^{-5}$, weight decay: $10^{-5}$, $r_{\text{demo}}$: 0.01, batch size: 8, number of attention layer: 8, number of head of attention layer: 4, window size $w = 21$, dropout: 0.1, buffer size: 64, training episode: 2000.

- **FSAF:** alpha: 0.8, hidden size: 100, learning rate: 0.01, batch size: 128, few shot step: 5, number of particles: 5, total task: 3, size of meta data: 100, use demo: True, early terminate: False, select type: average, training episode: 300 for $K = 2$ and 500 for $K = 3$.

- **OptFormer:** string length: 128, learning rate: $10^{-2}$, weight decay: $10^{-2}$, batch size: 1, window size $w = 10$, training episode: 1000.

- **Common Hyperparameter:** optimizer: Adam (Kingma & Ba, 2015), $\epsilon$-greedy rate: 0.1

- **Transformer Architecture:** GPT-2-based Transformer architecture.

### A.3 Hyperparameters of Rule-Based Approaches

- **qNEHVI:** q(batch selection): 1

- **qHVKG:** q(batch selection): 1

- **qParEgo:** q(batch selection): 1, acquisition function: Expected Improvement

- **JES:** # of samples: 64, estimation type: LB

- **NSGA2:** population size: 10, # of generations: 10, sampling: random

## B  Additional Related Work

### B.1  Transformers and Sequence Modeling for RL

Given the success of sequence-to-sequence models in language processing, RL has recently been addressed through the lens of sequence modeling, especially transformers. For example, Chen et al. (2021) proposed Decision Transformer (DT), which addresses offline RL by mapping return-to-go to actions via transformers and thereby substantiating the concept of Upside Down RL (Schmidhuber, 2019). The design of DT has been subsequently extended to various settings, including online RL (Zheng et al., 2022), multi-game setting (Lee et al., 2022), and general information matching (Furuta et al., 2021). Concurrently, to tackle offline RL, Janner et al. (2021) propose Trajectory Transformer (TT), which serves as a predictive dynamics model and uses beam search as a trajectory optimizer. More recently, Chebotar et al. (2023) introduced Q-Transformer (QT), which is trained by offline temporal difference updates and achieves scalable representation for Q-functions by discretizing action dimensions as tokens on a Transformer, with a focus on robotic tasks. By contrast, the proposed BOFormer is built on the (online) generalized DQN framework and designed for addressing MOBO based on multiple enhancements.

### B.2  A More Detailed Comparison to Q-Transformer

Q-Transformer (QT) (Chebotar et al., 2023) similarly incorporates the idea of utilizing a Transformer in learning the $Q$-function. Despite this high-level design resemblance, the proposed BOFormer is different from QT in multiple aspects:

- **Problem Setting:** QT is designed mainly to address offline RL, where the performance is highly correlated to the quality of prior data, as a robotic learning approach and required to address

---

[*]We reuse the open source implementation from https://github.com/jannerm/trajectory-transformer.

high-dimensional action spaces. By contrast, our work introduces BOFormer, which is trained to solve MOBO by learning a generalized Q function based on the online interactions with synthetic GP functions, and hence this can be viewed as an instance of online RL.

- **Network Architecture:** QT uses state-action sequences as the input of the Transformer. By contrast, to resolve the hypervolume identifiability issue and achieve cross-domain transferability simultaneously, the proposed transformer of BOFormer implements a generalized DQN and uses the $Q$-augmented observation representation as the input of the transformer. This approach can better address the non-Markovian property in MOBO.

- **Training Algorithm:** Compared to QT, the proposed BOFormer incorporates multiple practical enhancements, including $Q$-augmented representation, reward normalization, and demo policy and prioritized trajectory replay buffer for off-policy learning.

## C PSEUDO CODE AND ADDITIONAL IMPLEMENTATION DETAILS

### C.1 PSEUDO CODE OF BOFORMER

The detailed pseudo code of the training processes for BOFormer under off-policy learning and on-policy learning setting are provided in Algorithms 1 and 2, respectively.

### C.2 ADDITIONAL IMPLEMENTATION DETAILS OF BOFORMER

**Reward Signal With Normalization.** In the MOBO setting, one natural reward design is the one-step improvement in hypervolume, *i.e.,* $\hat{r}_t := \mathrm{HV}(\mathcal{X}_t) - \mathrm{HV}(\mathcal{X}_{t-1})$. However, as the achieved hypervolume increases, the reward signal $\hat{r}_t$ can get weaker in the later stage of an episode, making it difficult to recover the whole Pareto front. To address this, we construct $r_t$, which is $\hat{r}_t$ but scaled by the difference between the current hypervolume and optimal hypervolume, as the reward signal for RL in MOBO, *i.e.,*

$$r_t := \frac{\mathrm{HV}(\mathcal{X}_t) - \mathrm{HV}(\mathcal{X}_{t-1})}{\mathrm{HV}(\mathcal{X}^*) - \mathrm{HV}(\mathcal{X}_t)}. \tag{7}$$

**Remark C.1.** The information about $\mathrm{HV}(\mathcal{X}^*)$ in (7) is used only during training and can be easily pre-computed or approximated given the knowledge about the domain and the black-box functions.

**Demo-Policy-Guided Exploration.** To facilitate the off-policy learning of BOFormer, one natural approach is to adopt a behavior policy with randomized exploration (*e.g.,* epsilon-greedy) for collecting trajectories from the environment. However, such a randomized exploration scheme can be very inefficient as the sampling budget in MOBO is usually much smaller than the domain size (*i.e.,* the number of actions). To better guide the exploration, we propose to use a demo policy, which is possibly sub-optimal but of sufficient strength in exploring regions near the Pareto front. In practice, we set $r_{\mathrm{demo}}$ to be the probability of using a demo policy induced by an off-the-shelf AF for MOBO (*e.g.,* EHVI) in this training episode. The training processes of BOFormer for off-policy learning and on-policy learning are provided in Algorithms 1 and 2, respectively.

**Transfer Learning Across Different Numbers of Objective Functions.** As discussed in the main text, the input dimension of BOFormer varies depending on the number of objective functions. For instance, a BOFormer model trained on 2 objective functions cannot be directly applied to 3 objective functions due to the mismatch in input dimensions, which requires reshaping of the embedding layer. To address this issue and reduce the computational burden of training BOFormer, we developed an efficient transfer learning strategy that enables model transfering across different numbers of objective functions.

Our approach first loads the pre-trained weights for the transformer component from another trained BOFormer, then initializes a linear embedding layer to match the dimension of the state-action pair. Only the linear embedding layer is trained. This strategy significantly reduces training episodes and computational requirements by allowing us to leverage pre-trained models for different numbers of objective functions with minimal adjustments.

The efficiency of this transfer learning method is evident in the reduced number of training episodes. While the original model requires approximately 3000 episodes to converge, the transfer model

achieves comparable performance after only 400 episodes of fine-tuning. This substantial reduction in training episodes demonstrates the effectiveness of our approach in rapidly adapting pre-trained models to new objective function configurations.

**Positional Encoding.** The positional encoding in BOFormer is designed specifically for the context of sequential modeling in RL. Unlike the standard positional embeddings used in traditional Transformers, where each position corresponds to a single token, our method assigns an embedding to each timestep $t$, that is shared across multiple tokens (*i.e.,* $\{(s_t, a_t), r_t, Q(s_t, a_t)\}$ for state-action pairs, rewards, and Q-values). This approach allows the Transformer to capture the temporal context effectively by providing explicit time indices for each token.

**Temporal Information in State-Action Representation.** Temporal information is also incorporated into the state-action representation, specifically through terms like $j/t$ in (5). As BOFormer is an RL-based method, this essentially corresponds to the episodic RL setting (Dann et al., 2017; Neu & Pike-Burke, 2020), where the length of each episode is fixed. In episodic RL, the value functions $Q_t^{\pi}(s, a)$ and $V_t^{\pi}(s, a)$ do depend on this temporal information $t$ because the RL agent shall make decisions based on the remaining time budget (*i.e.,* $T - t$). In the context of MOBO, this temporal information also serves as a budget indicator, similar to that in episodic RL. This is critical for learning MOBO policies that can achieve high hypervolume in as few queries as possible by maximizing cumulative reward within the given budget. In summary, the motivations for these two temporal components differ:

- Positional Encoding ensures the Transformer can capture temporal information in the historical queries.

- State-Action Temporal Representation directly informs the RL agent of the progress within sequential decision-making with respect to the remaining budget, helping maximize future cumulative rewards.

Regarding the ablation studies for validating the effects of positional encoding and temporal information, we conduct a preliminary ablation study on BOFormer by comparing several variants of BOFormer models:

1. With Positional Encoding and Temporal Information: This is the original BOFormer.

2. Without Temporal Information: Temporal information is removed from BOFormer during training.

3. Without Positional Encoding nor Temporal Information: Both components were excluded from BOFormer during training.

To better observe the training behavior, we collected all training data using the demo policy and monitored the evolution of the loss curve during training. The rest of the configuration, such as hyperparameters and neural network architecture, was consistent across all three variants. The evolution of the training loss for the three variants can be found in Figure 5 We observe both second and third BOFormer appear to suffer from divergence in terms of training loss. These results appear to align with the requirements of episodic RL, where knowing the budget is important for estimating Q-values. Temporal information plays a vital role in the need of this budget-related representation, while positional encoding is essential for enabling the Transformer to recognize each position as corresponding to a single token. Together, these components are helpful to BOFormer's stability and convergence during training.

**Off-Policy Learning and Prioritized Trajectory Replay Buffer** These components are introduced to address exploration, sample efficiency and learning stability:

- Off-Policy Learning enables BOFormer to leverage previously collected data, reducing the need for additional costly sampling. This improves sample efficiency and facilitates the use of a demo policy, allowing for more effective learning and faster convergence.

- Prioritized Trajectory Replay Buffer ensures that the most informative transitions (*e.g.,* those with significant temporal-difference errors) are replayed more frequently during training, accelerating convergence.

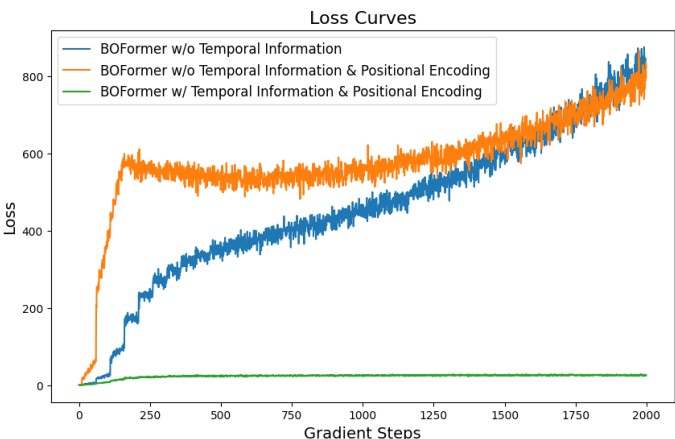

Figure 5: Training loss for $3$ variants of BOFormer models

**Handling Continuous Domains Under BOFormer.** As BOFormer is a learning-based acquisition function (AF) implemented as a neural network, it can nicely handle continuous domains by leveraging the common ways of finding an (approximate) AF maximizer, just like the existing learning-based methods and neural AFs. Specifically:

- Approximate maximization by Sobol grids (commonly used by learning-based methods): One can find an approximate maximizer of each learning-based AF by grid search with the help of Sobol sequences, as suggested by (Volpp et al., 2020; Hsieh et al., 2021). Specifically, one can (i) first selects $N$ points spanning over the entire domain with the help of Sobol sequence, (ii) chooses the top-$M$ points in terms of AF value among the $N$ candidates, (iii) and then builds a local Sobol grid of $K$ points for each of these $M$ points. This is the default choice of BOFormer.

- Gradient-based optimization with automatic differentiation (used by the rule-based methods in BoTorch): One can also optimize the AF by using gradient-based optimization (*e.g.,* multi-start L-BFGS-B in BOTorch available at `https://botorch.org/docs/optimization`, with the help of automatic differentiation provided by deep learning frameworks (*e.g.,* PyTorch). This is the default approach adopted by BoTorch.

**Q-Value Computation by Target Network for a Trajectory.** Given a trajectory $\tau = \{s_1, a_1, r_1, s_2, \cdots, s_{t-1}, a_{t-1}, r_{t-1}\}$: (i) BOFormer computes $Q(h_0, s_1, a_1)$ using $(s_1, a_1)$ as the input. (ii) Then, it computes $Q(h_1, s_2, a_2)$ by letting $(s_1, a_1, r_1, s_2, a_2)$ as the input. (iii) This process continues sequentially until $Q(h_{t-2}, s_{t-1}, a_{t-1})$ is computed.

**Action Selection by Policy Network at time $t$.** To select $a_t$ at time $t$, BOFormer samples actions from a softmax policy where the logits correspond to Q-values: $\Pr(a|h_{t-1}, s_t) = \exp\Big(Q(h_{t-1}, s_t, a)\Big)/\sum_{a' \in \mathcal{X}} \exp\Big(Q(h_{t-1}, s_t, a')\Big)$.

**Other Training Details.**

- Demo policy: Each trajectory is collected using the demo policy with a probability of $r_{\text{demo}}$; otherwise, it is collected by BOFormer itself.

- Batch sampling: BOFormer samples a batch from the prioritized trajectory experience replay buffer, with each batch containing a batch of trajectories.

- Optimization: Gradient descent is performed on the policy network using the optimizer with dropout.

- Target network: The target network is used to compute $\{Q(h_{i-1}, s_i, a_i)\}_{i=1}^{t-1}$, and synchronized with the policy network every $5$ episodes.

## C.3  ABOUT TEMPORAL INFORMATION IN THE HISTORY REPRESENTATION OF BOFORMER

As BOFormer is a non-Markovian RL method for BO, and hence the design shall take both RL and BO into account. Specifically:

- **Temporal information in state-action representation serves as a budget indicator, similar to episodic RL**: In MOBO, we do want to attain high hypervolume by using as few samples as possible, and hence each episode only has a small number of queries (or time steps), *e.g.,* $T = 100$. As BOFormer is an RL-based method, this essentially corresponds to the episodic RL setting (*e.g.,* [Dann et al., 2017; Neu and Pike-Burke, 2020]), where the length of each episode is fixed. In episodic RL, the value functions $Q_t^\pi(s, a)$ and $V_t^\pi(s, a)$ do depend on this temporal information $t$ because the RL agent shall make decisions based on the remaining time budget (*i.e.,* $T - t$). In the context of MOBO, this temporal information also serves as a budget indicator, similar to that in episodic RL.

- **Permutation-invariance of queries in posterior distributions**: Anothe notable fact is that in BO, the historical queries shall be permutation-invariant in the sense that the order of previous queries should not affect the posterior distributions under GP.

By taking both into account, we choose to include the temporal information in the state-action representation.

## C.4  ISSUES WITH THE DIRECT IMPLEMENTATION IN SECTION 4.1

Recall that the direct implementation in Section 4.1 is designed to implement Generalized DQN with the per-step observation that consists of two parts:

1. The posterior distributions of all $K$ objective functions at all the domain points.

2. The best function values observed so far (denoted by $y^{(i)*}$, $i = 1, ..., K$).

Hence, the per-step observation is $\{(\mu^{(i)}(x), \sigma^{(i)}(x), y^{(i)*})_{x \in \mathbb{X}, i \in [K]}\}$. As described in Section 4.1, this representation design is subject to two practical issues:

- **Scalability issue in sequence length:** Under this design, the sequence length of the per-step observation would grow linearly with the number of domain points and pose a stringent requirement on training.

- **Limited cross-domain transferability:** As this observation representation is domain-dependent, the learned model is tightly coupled with the training domain and has very limited transferability.

We have tried this direct implementation, and we found that this design severely suffers from the scalability issue and requires extremely long training time. Below we report the training time that we observed:

- Even under a fairly small input domain of only 50 points (*i.e.,* $|\mathbb{X}| = 50$), we observe that 20 training iterations already take more than 36 hours of wall clock time to complete. **To finish at least 1000 training iterations (based on our training experience with BOFormer), this direct implementation would require about 75 days to complete one training run.

- Notably, the above training was already on a high-end GPU server with NVIDIA RTX 6000 Ada Generation GPUs and Intel Xeon Gold 5515+ CPU.

The above issue will be more severe under an input domain with more domain points (*i.e.,* larger $|\mathbb{X}|$). This manifests that this direct implementation suffers significantly from the scalability issue and is rather impractical. This also motivates the proposed BOFormer for substantiating the Generalized DQN framework.

## C.5   Implementation Details of OptFormer

Recall that the OptFormer and BoFormer have different application scopes: BOFormer is designed to serve as a general-purpose optimization solver for MO black-box optimization, whereas OptFormer is more task-specific as it is designed specifically for single-objective HPO and requires offline HPO data for training. To adapt OptFormer to the general MOBO setting, there are some modifications needed: (i) The core idea of OptFormer is to recast HPO as language modeling, and we adapt this idea to MOBO. For instance, when training OptFormer for a 2-objective MOBOsetting, we modify the input to include a representation that is more relevant to the MOBO problem, such as

$$\text{``1st\_dimension\_x} = 0.031 \quad \text{2nd\_dimension\_x} = 0.531$$
$$\text{1st\_objective\_y} = 0.254 \quad \text{2nd\_objective\_y} = 0.258$$
$$\text{1st\_dimension\_x} = 0.604 \cdots \text{''} \tag{8}$$

To ensure the strength of OptFormer, we retrain the OptFormer model upon handling different numbers of objective functions. (ii) Additionally, the metadata includes information derived from the GP surrogate model, which is estimated online by maximizing the marginal likelihood (just like all the other rule-based AFs and BOFormer). Given that OptFormer requires an offline dataset for training, we collected the training dataset using rule-based AFs (*e.g.,* Expected Hypervolume Improvement), to ensure the model is effectively tailored to the MOBO setting.

# D   Detailed Experimental Results

## D.1   Detailed Hypervolume Statistics and Inference time

In this section, we provide the additional and detailed statistics of the per-sample inference time and the final hypervolume as follows.

Table 3: Average per-sample inference time of BOFormer and other benchmark methods.

| Algorithms | Time (millisecond) | | |
|---|---|---|---|
| | 2 Objectives | 3 Objectives | 4 Objectives |
| BOFormer | 14.20 | 14.20 | 14.29 |
| qNEHVI | 6.15 | 8.042 | 224.10 |
| JES | 46.66 | 50.15 | 57.87 |
| FSAF | 5.49 | 6.55 | 6.92 |
| DT | 1.30 | 1.28 | 1.28 |
| NSGA2 | 0.17 | 0.19 | 0.20 |
| qParEGO | 1.10 | 1.42 | 1.75 |
| OptFormer | 237.33 | 366.15 | 485.51 |
| qHVKG | 371.66 | 533.37 | 662.76 |
| QT | 9.37 | 9.71 | 9.73 |

The messages from Table 3 are mainly three-fold:

- We observe that for the baselines that are competitive in the attained hypervolume (such as qNEHVI and qHVKG), the inference time scales significantly with the number of objectives, whereas BOFormer maintains a lower and nearly constant inference time across different number of objectives.

- Compared to the Markovian approach like FSAF, BOFormer only needs a slightly higher per-step inference time and remains rather efficient. Hence, the non-Markovian nature of BOFormer still consumes an inference time comparable to a Markovian approach.

- BOFormer enjoys a much lower inference time than the sophisticated OptFormer, which views black-box optimization as a language modeling problem in a text-to-text manner and hence involves text-based input representation.

## D.2   Ablation Study on the Demo Policy in BOFormer

---

**Algorithm 1** Off-Policy BOFormer

---

1: **Input:** $\boldsymbol{\theta}_1$, Training Episodes $E$, Time Horizon $T$, Demo Rate $r_{\text{demo}}$, Target Rate $r_{\text{target}}$, Target Network $Q_{\bar{\theta}}$, Batch Size $B$, Replay buffer $\mathcal{B}$, environment $\mathcal{E}$.
2: **for** $e = 1, 2, \cdots, E$ **do**
3:    $\mathcal{E}$.reset()
4:    $s_1^e = \mathcal{E}$.state
5:    demo = random.binomial($r_{\text{demo}}$)
6:    $\pi_t^e$ = ExpectedHypervolumeImprovement
7:    **for** $t = 1, 2, \cdots, T$ **do**
8:      **if** demo **then**
9:        $x_t^e = \pi_t^e(s_t^e)$
10:      **else**
11:        **for** $i = 1, 2, \cdots, t-1$ **do**
12:          Compute $Q_{\bar{\theta}}(h_i^e, o_i^e(x_i^e)) = Q_{\bar{\theta}}\left(\left\{o_j^e(x_j^e), r_j^e, Q_{\bar{\theta}}(h_{j-1}^e, o_{j-1}^e(x_{j-1}^e))\right\}_{j=1}^{i-1}, o_i^e(x_i^e)\right).$
13:        **end for**
14:        Select $x_t^e := \operatorname{argmax}_{x \in \mathbb{X}} Q_{\hat{\theta}}(h_t^e, o_t^e(x))$
15:      **end if**
16:      $s_{t+1}^e, r_t^e = \mathcal{E}$.step($x_t^e$)
17:    **end for**
18:    **if** $e \bmod r_{\text{target}} == 0$ **then**
19:      $Q_{\bar{\theta}} = Q_{\boldsymbol{\theta}_e}$
20:    **end if**
21:    $\mathcal{B}$.append($\tau_e = \{o_i^e(x_i^e), r_i^e\}_{i=1}^T$)
22:    Sample B batches from $\mathcal{B}$.
23:    **for** $b = 1, 2, \cdots, B$ **do**
24:      Compute $Q_{\hat{\theta}}^b(h_1^b, o_1^b(x_1^b)) = Q_{\bar{\theta}}\left(o_1^b(x_1^b)\right)$
25:      **for** $i = 1, 2, \cdots, T-1$ **do**
26:        Compute $Q_{\hat{\theta}}^b(h_i^b, o_i^b(x_i^b)) = Q_{\hat{\theta}}\left(\left\{o_i^b(x_j^b), r_j^b, Q_{\bar{\theta}}(h_{j-1}^b, o_{j-1}^b(x_{j-1}^b))\right\}_{j=1}^{i-1}, o_i^b(x_i^b)\right).$
27:        Compute $Q_{\bar{\theta}}^b(h_i^b, o_i^b(x_i^b)) = Q_{\bar{\theta}}\left(\left\{o_j^b(x_j^b), r_j^b, Q_{\bar{\theta}}(h_{j-1}^b, o_{j-1}^b(x_{j-1}^b))\right\}_{j=1}^{i-1}, o_i^b(x_i^b)\right).$
28:      **end for**
29:    **end for**
30:    $\text{Loss}(\theta) = \sum_{b=1}^B \sum_{t=1}^{T-1} \left( Q_{\hat{\theta}}^b(h_i^b, o_i^b(x_i^b)) - (r_t^b + \max_{x \in \mathbb{X}} Q_{\bar{\theta}}^b(h_i^b, o_i^b(x))) \right)^2$
31:    $\boldsymbol{\theta}_{e+1} = \underset{\boldsymbol{\theta}}{\operatorname{argmin}}\, \text{Loss}(\theta)$
32: **end for**

---

**Algorithm 2** On-Policy BOFormer

---

1: **Input:** $\boldsymbol{\theta}_1$, Training Episodes $E$, Time Horizon $T$, environment $\mathcal{E}$,
2: **for** $e = 1, 2, \cdots, E$ **do**
3:    $\mathcal{E}$.reset()
4:    $s_1 = \mathcal{E}$.state
5:    **for** $t = 1, 2, \cdots, T$ **do**
6:      Select $x_t := \operatorname{argmax}_{x \in \mathbb{X}} Q_{\theta_t}(h_t, o_t(x))$
7:      $o_{t+1}(x), r_t = \mathcal{E}$.step($x_t$)
8:      $\theta_{t+1} = \underset{\boldsymbol{\theta}}{\operatorname{argmin}} \left( Q_\theta(h_t, o_t(x_t)) - (r_t + \gamma \max_{x \in \mathbb{X}} Q_{\boldsymbol{\theta}_t}(h_{t+1}, o_{t+1}(x))) \right)^2$
9:    **end for**
10: **end for**

---

To further investigate the connection between BOFormer's performance and the choice of demo policy, we conducted additional experiments comparing qNEHVI (the original choice) and NSGA2 as demo policies, as well as a baseline where no demo policy is used.

The results are available at Figure 10. Based on these results, we can observe the following:

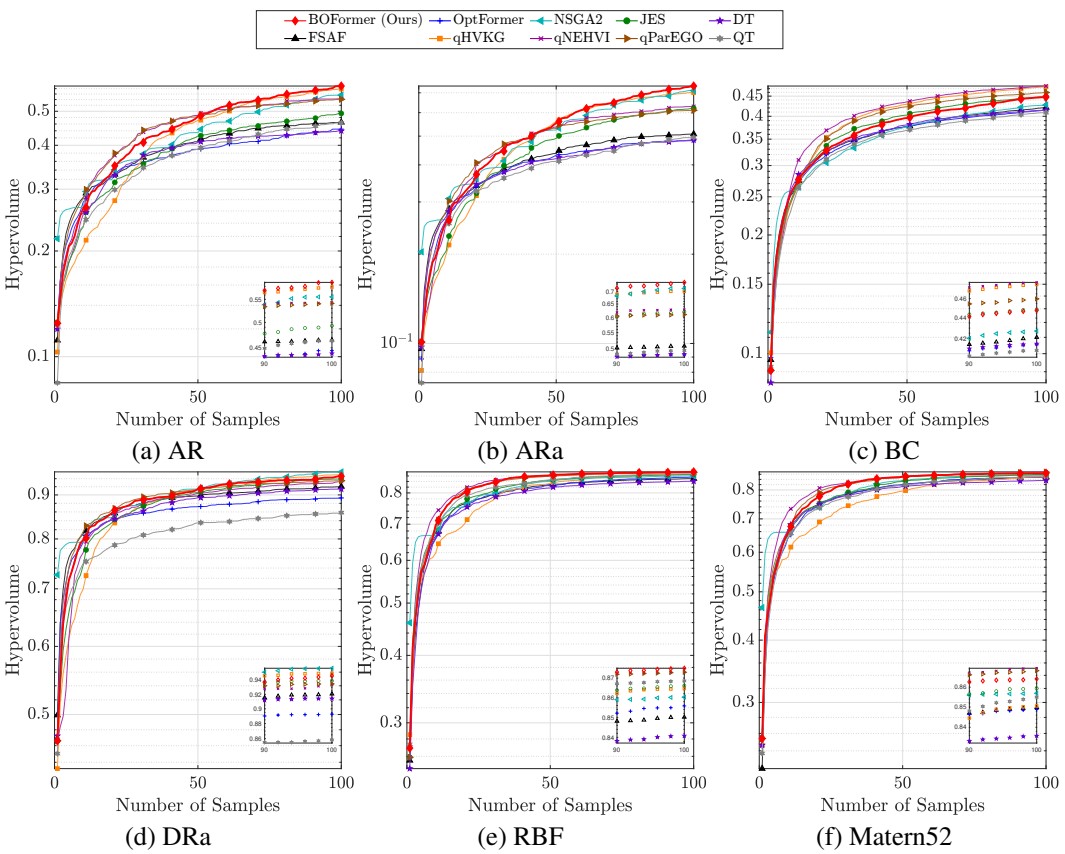

Figure 6: Averaged attained hypervolume under synthetic functions with two objectives.

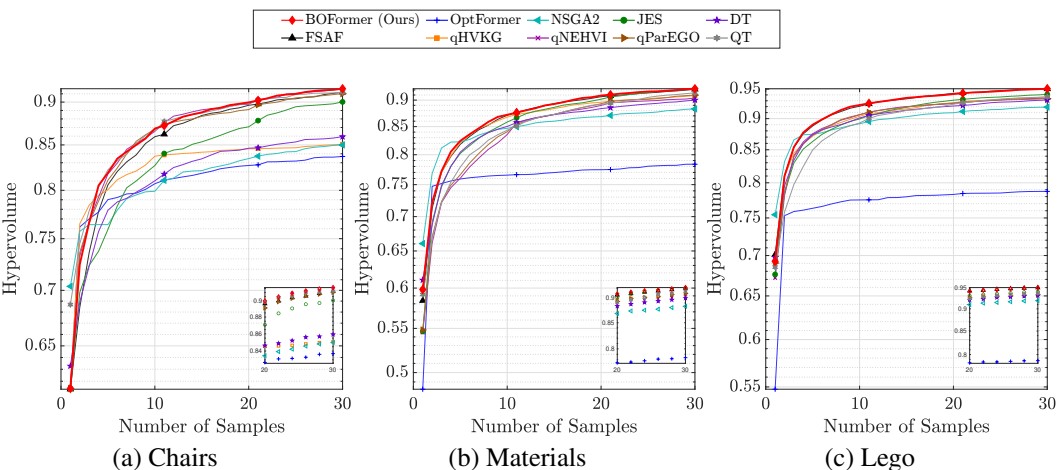

Figure 7: Averaged attained hypervolume under HPO-3DGS with two objectives.

- BOFormer (w/i qNEHVI) achieves favorable improvement in attained hypervolume than that without using a demo policy. This demonstrates the benefit of off-policy learning via a demo policy in the context of MOBO. By contrast, the improvement offered by the NSGA2-based demo policy appears minimal.

- Recall that qNEHVI performs generally better than NSGA2 on these tasks (*i.e.,* RBF, Matern, and BC) as shown in Table 1 of the original manuscript and the performance profiles in Figure 3. Intuitively, we would expect that the trajectories contributed by qNEHVI can

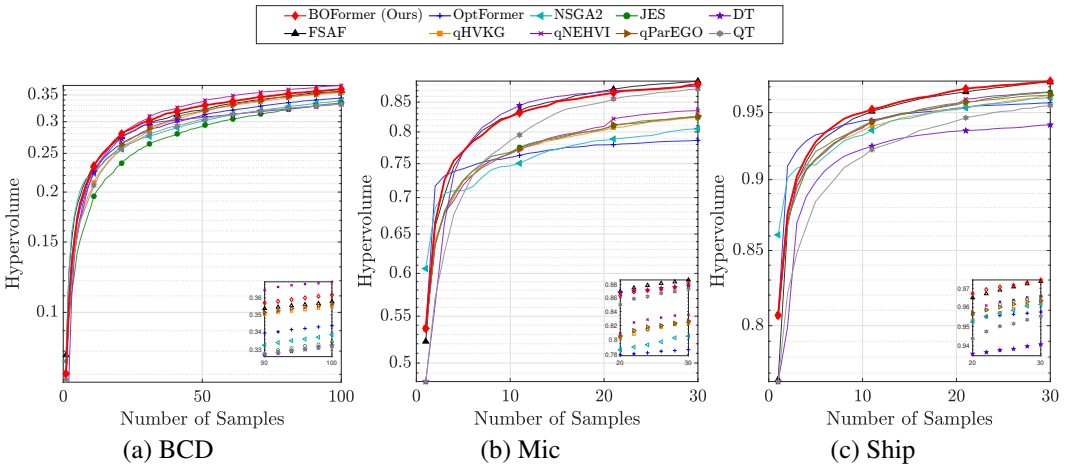

Figure 8: Averaged attained hypervolume under synthetic and HPO-3DGS with three objectives.

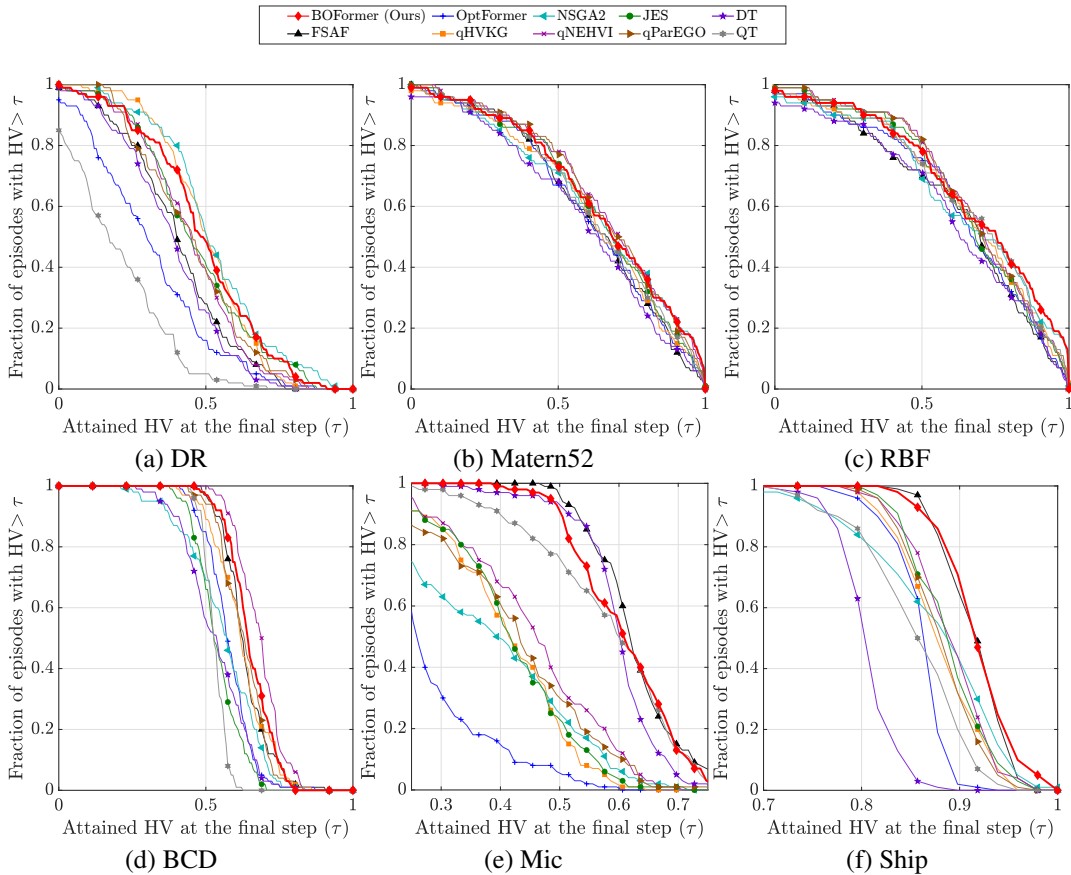

Figure 9: Performance profiles in terms of hypervolume at the final step under synthetic functions and HPO-3DGS.

better help BOFormer explore the regions with higher hypervolume. This intuition also resonates with the general understanding that RL performance can be correlated with the data quality (Kumar et al., 2019).

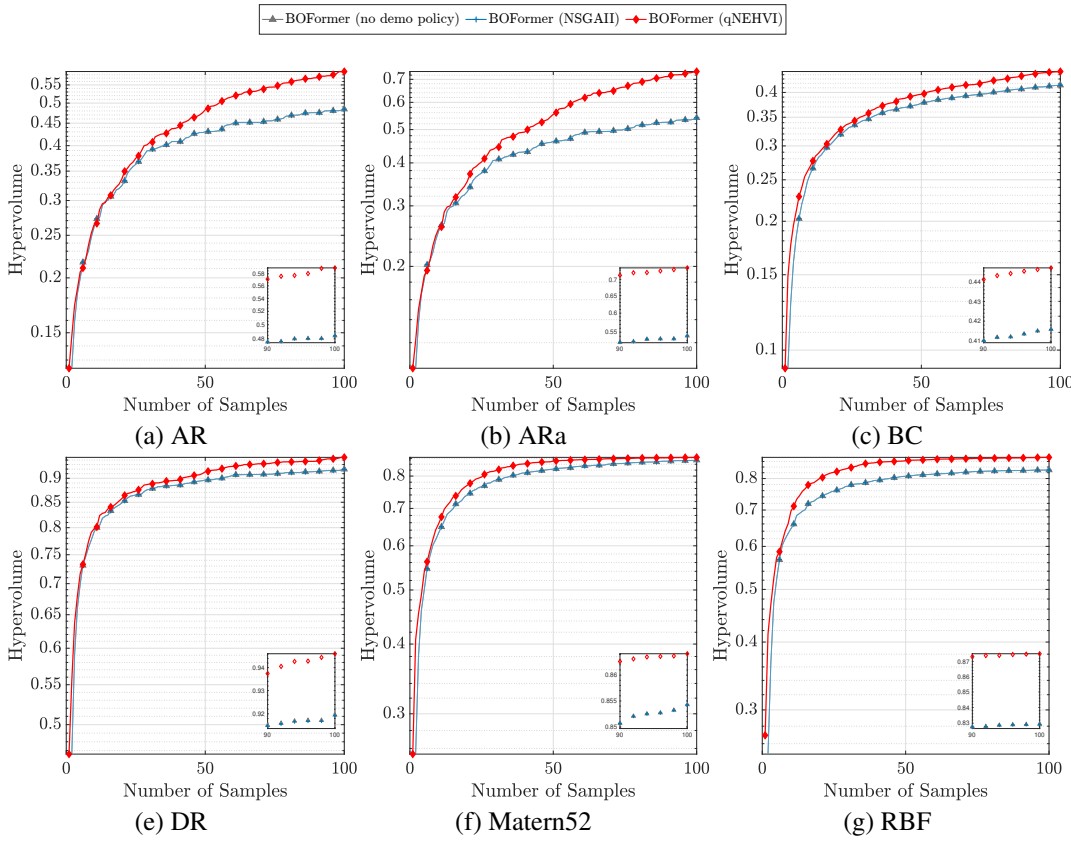

Figure 10: Averaged attained hypervolume under synthetic functions with two objectives.

### D.3 SCALABILITY OF BOFORMER TO HIGH-DIMENSIONAL PROBLEMS

We conducted additional experiments on black-box functions with higher-dimensional domains, including:

- Matern52 and RBF functions under $d = 10$ and $d = 30$.

- (Ackley & Rosenbrock) and (Ackley & Rastrigin) under $d = 40$.

- DTLZ functions under $d = 100$. Notably, DTLZ is a family of synthetic functions for MOBO and included in the pymoo library.

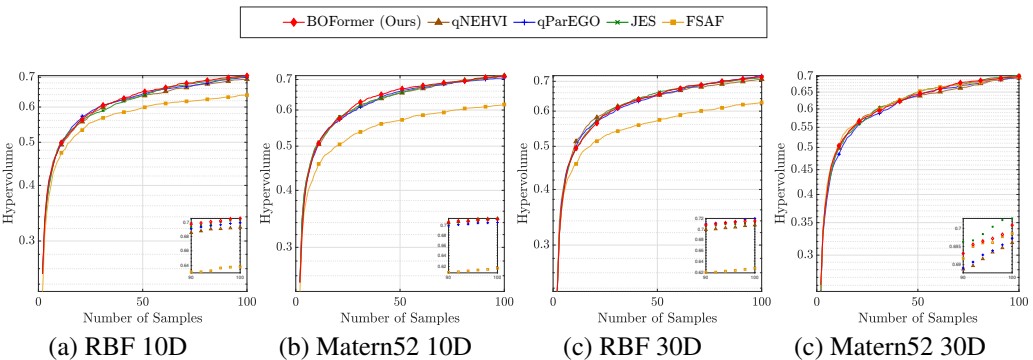

Figure 11: Averaged attained hypervolume under more challenging GP functions with higher-dimensional domains.

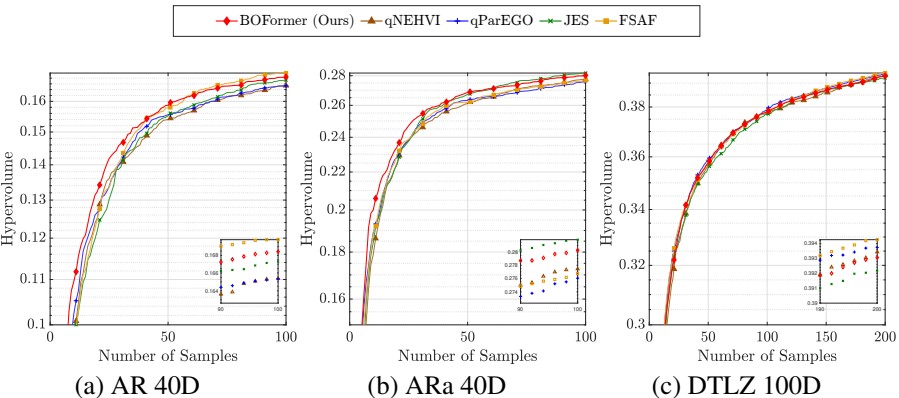

Figure 12: Averaged attained hypervolume under various challenging functions, including synthetic functions with 40-dimensional domains and the DTLZ function with a 100-dimensional domain.

Figures 11-12 demonstrate that BOFormer remains competitive compared to other MOBO baseline algorithms on these high-dimensional tasks. while FSAF requires metadata for few-shot updates,. Notably, all the above results are obtained under the same BOFormer model without any fine-tuning, and hence this further demonstrates the strong cross-domain transferability of BOFormer.

### D.4 Performance of BOFormer on More Challenging Problems

We conducted additional experiments on various challenging problems with more than 100 samples, including scenarios with (1) higher domain dimensionality and (2) increased perturbation noise. Accordingly, we set $T = 200$ or all the following experiments to better evaluate BOFormer and other baselines. Figure 13 demonstrates that BOFormer remains competitive compared to other MOBO baseline methods across various types of challenging scenarios. These findings highlight BOFormer's scalability, making it well-suited for more challenging black-box functions.

**Remark D.1.** To make the problems more challenging as suggested, for the HPO-3DGS tasks, the perturbation noise is set as 0.1 to make the black-box functions more non-smooth.

**Remark D.2.** In the above, we compare BOFormer with the rule-based methods such as qNEHVI, JES, and qParEGO as well as the learning-based method like FSAF. These MOBO methods have been shown to be the most competitive among all the baselines (cf. Tables 1-2)

**Remark D.3.** FSAF is a metaRL-based method and requires some metadata for few-shot adaptation. By contrast, BOFormer is evaluated on unseen testing functions in a zero-shot manner.

## E About the Non-Markovian Nature of MOBO

Defining the state representation as the complete set of historical queries along with the posterior distribution of candidate points would indeed endow the problem with *full observability*. However, by the standard definition of Markov property, this design is not Markovian (and hence does not result in an MDP) because the definitions of the Markov property and MDP presume that the state space is of fixed dimensionality. Specifically, this design involves two fundamental issues:

- **The dimensionality of the state space would increase across time steps as more samples are taken**: To illustrate this more concretely, let us consider an example of $K$-objective MOBO problem with objective functions $\mathbf{f}(x) = (f_1(x), ..., f_K(x))$ with $\mathbf{f}(x) : \mathbb{X} \to \mathbb{R}^K$, where the domain $\mathbb{X}$ is discrete and has $N$ candidate points (a similar argument can be used for continuous domains). Recall that we use $(x_i, \mathbf{y}_i)$ to denote the query and the resulting observed function values at time step $i$. In this design, at the $t$-th step of an episode, the state representation would consist of: (i) The complete set of historical queries

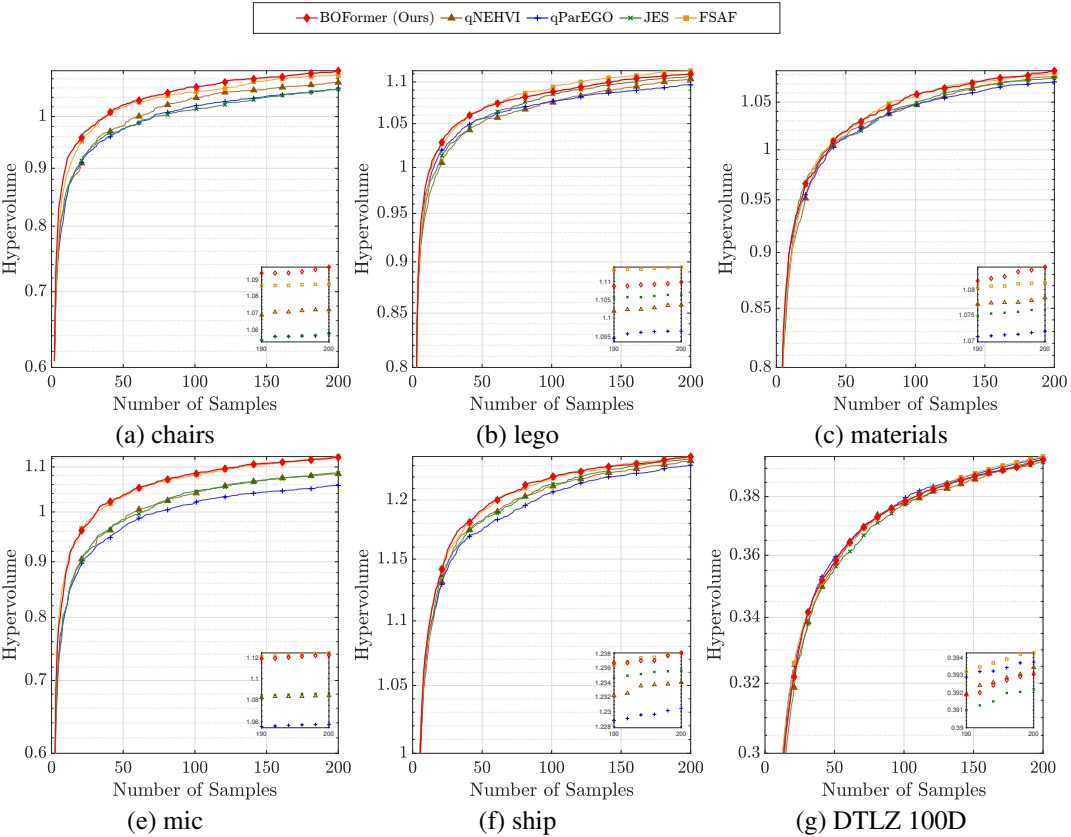

Figure 13: Averaged attained hypervolume under more challenging problems with 200 samples

is $\{(x_1, \mathbf{y}_1), \ldots, (x_t, \mathbf{y}_t)\}$. The dimensionality of this part is $t(K + 1)$. (ii) The posterior distribution of candidate points can be captured by $\{\mu_t(x), \sigma_t(x)\}_{x \in X}$. The dimensionality of this part is $2KN$. Therefore, we can see that the dimensionality of the state space would increase with time step $t$.

- **The state representation would be tightly coupled with the input domain**: Based on the above first point, we can see that under this design, part of the state representation involves $\{(x_1, \mathbf{y}_1), \ldots, (x_t, \mathbf{y}_t)\}$. This suggests that the state space is coupled with the domain of each specific task. As a result, re-training is typically needed upon handling a new task. This issue motivates the need and design for BOFormer, which has the favorable cross-domain capability (i.e., the size and the dimensionality of the input domains can be different between the training and testing).

