# OpenReview forum: "BOFormer: Learning to Solve Multi-Objective Bayesian Optimization via Non-Markovian RL"
_ICLR.cc/2025/Conference — ICLR 2025 Poster_

### Official Review · Reviewer_UwND · 2024-10-30

**Soundness:** 3
**Presentation:** 2
**Contribution:** 2
**Rating:** 8
**Confidence:** 3

**Summary:**

This paper introduces BOFormer, a learning-based approach for Multi-Objective Bayesian Optimization (MOBO). The key idea is to replace the traditional handcrafted acquisition function with a Transformer-based network trained through non-Markovian reinforcement learning. The authors claim that MOBO is inherently non-Markovian due to the "hypervolume identifiability issue" and propose a generalized DQN framework incorporating historical information. The method is evaluated on both synthetic functions and real-world hyperparameter optimization tasks.

**Strengths:**

* While there are several existing works on meta-learned acquisition functions for single-objective BO, BOFormer is the first work to apply a learning-based acquisition function to MOBO.
* The domain-agnostic representation, which only incorporates historical query information rather than maintaining information for all domain points, enables cross-domain transfer while being memory-efficient.
* Well-structured related work with clear categorization of existing approaches and insightful comparison with similar works, especially in distinguishing the unique aspects of BOFormer from OptFormer and NAP.
* The paper provides a comprehensive evaluation on various synthetic functions and real-world hyperparameter optimization tasks, both show strong performance of BOFormer. Besides, the paper conducts an additional ablation study on sequence length.

**Weaknesses:**

* I have some doubts about the non-Markovian nature of MOBO. If we define the state as the complete set of historical queries along with the posterior distribution of candidate points, the problem naturally becomes Markovian, as the hypervolume improvement can be uniquely determined from this state representation. The non-Markovian property in your approach appears to be artificially induced by your choice of a simplified state representation. Please correct me if I am wrong or if I misunderstood something, as I am not from the field of reinforcement learning.
* A major weakness lies in the methodology section, particularly in the insufficient details about BOFormer. Figure 2 is not thoroughly discussed; for instance, positional encoding appears in the architecture diagram but is never mentioned in the text, while Equation 5 already seems to contain temporal information. Moreover, it's unclear how j/t in Equation 5 is combined and encoded with other inputs, as it appears to be merely an arithmetic sequence (e.g., numbers from 0.1 to 0.9 when t=10). Additionally, the introduction of Off-Policy Learning and Prioritized Trajectory Replay Buffer feels abrupt - what problems are they addressing? There's no proper introduction or motivation. I suggest the authors improve this section to present the problems and their solutions more clearly to readers.
* Another concern is the temporal information in the history representation. In BO, the historical queries should be permutation-invariant - the order of previous queries should not affect future decisions. However, by incorporating explicit temporal information, the method might be breaking this consistency. Or have you observed any benefit of incorporating temporal information? And is there any ablation study showing the impact of it?
* The paper lacks some experimental details. E.g., no configuration of the surrogate model such as the choice of the kernel and hyperparameters; Training cost for meta-training not reported.
* The paper contains several mathematical formulation issues and inconsistencies. E.g., Line 345, shouldn't it be $max$?; Line 347, I guess it should be $i$-th objective as $j$ is the time step; Inconsistent index notation, in line 344, $i$ indexes objective functions: $i \in [1,···,K]$, while in the equation at line 374, $i$ represents time steps; Line 225, the hypervolume formula contains an unexplained $x'$ that appears redundant; Line 819, shouldn't it be uniform distribution?
* There is no discussion about Figure 3, it might be better to give an introduction about performance profiles.
* There is no experiment to support the idea of demo-policy-guided exploration, thus it is hard to see the effect of this enhancement.
* Minor issues: Line 201: Incorrect appendix reference (B.2 instead of B); Line 820: Incomplete sentence ending abruptly.

**Questions:**

* Regarding OptFormer Implementation, how did you adapt OptFormer for MOBO? To my understanding it was originally designed for single-objective HPO tasks. Did you retrain the model from scratch for MOBO or fine-tune the existing model? It would be great if you could add some details for the baselines you compared with.
* Have you tried to compare the direct implementation you mentioned in Section 4.1 with your proposed method?
* I assume BOFormer can only accept discrete inputs, how does the method handle continuous domains in practice? Are there any discretization issues?

---

> ### Author Response · Authors · 2024-11-26
> **Response to Reviewer UwND**
>
> We sincerely thank the reviewer for the insightful feedback. Below, we address the questions raised by the reviewer via the point-to-point response. We hope the response could help the reviewer further recognize our contributions. Thank you.
>
> ### **Q1: About the non-Markovian nature of MOBO: Can we make the problem Markovian by defining the state as the complete set of historical queries along with the posterior distribution of candidate points?**
>
> Defining the state representation as the complete set of historical queries along with the posterior distribution of candidate points would indeed endow the problem with **full observability**. However, by the standard definition of Markov property, this design is not Markovian (and hence does not result in an MDP) because the definitions of the Markov property and MDP presume that the state space is of fixed dimensionality. Specifically, this design involves two fundamental issues:
>
> **(1) The dimensionality of the state space would increase across time steps as more samples are taken**: To illustrate this more concretely, let us consider an example of $K$-objective MOBO problem with objective functions $\mathbf{f}(x)=(f_1(x),...,f_K(x))$ with $\mathbf{f}(x): \mathbb{X} \rightarrow \mathbb{R}^d$, where the domain $\mathbb{X}$ is discrete and has $N$ candidate points (a similar argument can be used for continuous domains). Recall that we use $(x_i, \mathbf{y}_i)$ to denote the query and the resulting observed function values at time step $i$. In this design, at the $t$-th step of an episode, the state representation would consist of:
> - The complete set of historical queries is $\{(x_1, \mathbf{y}_1), … ,(x_t, \mathbf{y}_t)\}$. The dimensionality of this part is $t(K+1)$.
> - The posterior distribution of candidate points can be captured by $\\{\mathbf{\mu}\_t(x), \mathbf{\sigma}\_t(x)\\}\_{x\in X}$. The dimensionality of this part is $2KN$.
>
> Therefore, we can see that the dimensionality of the state space would increase with time step $t$.
>
> **(2) The state representation would be tightly coupled with the input domain**: Based on (1), we can see that under this design, part of the state representation involves $\{(x_1, \mathbf{y}_1), … ,(x_t, \mathbf{y}_t)\}$. This suggests that the state space is coupled with the domain of each specific task. As a result, re-training is typically needed upon handling a new task. This issue motivates the need and design for BOFormer, which has the favorable cross-domain capability (i.e., the size and the dimensionality of the input domains can be different between the training and testing).
>
> ### **Q2: Implementation details about BOFormer (e.g., positional encoding and temporal information in state-action representation in Figure 2, Off-Policy Learning, and Prioritized Trajectory Replay Buffer)**
>
> Thank you for the suggestion. We describe the implementation in more detail as follows:
>
> **Positional Encoding:** The positional encoding in BOFormer is designed specifically for the context of sequential modeling in RL. Unlike the standard positional embeddings used in traditional Transformers, where each position corresponds to a single token, our method assigns an embedding to each timestep $t$, that is shared across multiple tokens (i.e.,
> $\\{(s_t,a_t),r_t,Q(s_t,a_t)\\}$ for state-action pairs, rewards, and Q-values). This approach allows the Transformer to capture the temporal context effectively by providing explicit time indices for each token.
>
> **Temporal Information in State-Action Representation:**
> Temporal information is also incorporated into the state-action representation, specifically through terms $j/t$ in Eq. (5). This mechanism serves a distinct role: it acts as a "budget indicator," allowing the RL agent to understand how much of the budget remains. This is critical for learning MOBO policies that can achieve high hypervolume in as few queries as possible by maximizing cumulative reward within the given budget.
>
> In summary, the motivations for these two temporal components differ:
> * Positional Encoding ensures the Transformer can capture temporal information in the **historical** queries.
> * State-Action Temporal Representation directly informs the RL agent of the progress within sequential decision-making with respect to the remaining budget, helping maximize **future** cumulative rewards.
>
> **Off-Policy Learning and Prioritized Trajectory Replay Buffer:**
> These components are introduced to address exploration, sample efficiency and learning stability:
>
> * Off-Policy Learning enables BOFormer to leverage previously collected data, reducing the need for additional costly sampling. This improves sample efficiency and facilitates the use of a demo policy, enabling more effective learning and faster convergence.
>
> * Prioritized Trajectory Replay Buffer ensures that the most informative transitions (e.g., those with significant temporal-difference errors) are replayed more frequently during training, accelerating convergence.

---

> ### Author Response · Authors · 2024-11-26
> **Response to Reviewer UwND**
>
> ### **Q3: More explanation on the performance profiles in Figure 3**
>
> The performance profiles in Figure 3 are meant to more reliably present the performance
> variability of BOFormer and other baselines across testing episodes than the interval estimates of aggregate metrics. Specifically, the performance profiles show the (empirical) score distributions, indicating the fraction of runs that achieve a score above a certain normalized threshold. The performance profile was originally introduced by [Dolan and Moré, 2002] for benchmarking optimization software and later recommended by [Agarwal et al., 2021] to deep RL.
>
> In the context of MOBO, Figure 3 shows the score distribution of the attained final hypervolume, and the more top-right a curve resides the better.
>
> Based on Figure 3, we observe that in most tasks, the profiles of BOFormer (i.e., the curves in red) sit on the top right of the other MOBO baselines and hence enjoy a statistically better performance in terms of hypervolume.
>
> [Agarwal et al., 2021] R. Agarwal, M. Schwarzer, P. S. Castro, A. C. Courville, and M. Bellemare, "Deep reinforcement learning at the edge of the statistical precipice," NeurIPS 2021.
>
> [Dolan and Moré, 2002] Elizabeth D Dolan and Jorge J Moré, “Benchmarking optimization software with performance profiles,” Mathematical Programming, 2002.
>
> ### **Q4: How did you adapt OptFormer originally designed for single-objective HPO to MOBO**
> Recall that the OptFormer and BoFormer have different application scopes: BOFormer is designed to serve as a general-purpose optimization solver for MO black-box optimization, whereas OptFormer is more task-specific as it is designed specifically for single-objective HPO and requires offline HPO data for training.
>
> To adapt OptFormer to the general MOBO setting, there are some modifications needed:
>
> (1) The core idea of OptFormer is to recast HPO as language modeling, and we adapt this idea to MOBO. For instance, when training OptFormer for a 2-objective MOBOsetting, we modify the input to include a representation that is more relevant to the MOBO problem, such as
>
> `1st_dimension_x=0.031 2nd_dimension_x=0.531 1st_objective_y=0.254 2nd_objective_y=0.258 1st_dimension_x=0.604…`
>
> To ensure the strength of OptFormer, we retrain the OptFormer model upon handling different numbers of objective functions.
>
> (2) Additionally, the metadata includes information derived from the GP surrogate model, which is estimated online by maximizing the marginal likelihood (just like all the other rule-based AFs and BOFormer). Given that OptFormer requires an offline dataset for training, we collected the training dataset using rule-based AFs (e.g., Expected Hypervolume Improvement), to ensure the model is effectively tailored to the MOBO setting.
>
> ### **Q5: About the temporal information in the history representation: In BO, the historical queries should be permutation-invariant - the order of previous queries should not affect future decisions. However, by incorporating explicit temporal information, the method might be breaking this consistency.**
>
> Thanks for the insightful question. We would like to first highlight that BOFormer is a non-Markovian RL method for BO, and hence the design shall take both RL and BO into account.
>
> - **Temporal information in state-action representation serves as a budget indicator, similar to episodic RL**: In MOBO, we do want to attain high hypervolume by using as few samples as possible, and hence each episode only has a small number of queries (or time steps), e.g., $T=100$. As BOFormer is an RL-based method, this essentially corresponds to the episodic RL setting (e.g., [Dann et al., 2017; Neu and Pike-Burke, 2020]), where the length of each episode is fixed. In episodic RL, the value functions $Q^{\pi}_t(s,a)$ and $V^{\pi}_t(s,a)$ do depend on this temporal information $t$ because the RL agent shall make decisions based on the remaining time budget (i.e., $T-t$). In the context of MOBO, this temporal information also serves as a budget indicator, similar to that in episodic RL.
>
> - **Permutation-invariance of queries in posterior distributions**: We agree with the reviewer in that in BO, the historical queries shall be permutation-invariant in the sense that the order of previous queries should not affect the posterior distributions under GP.
>
> By taking both into account, we choose to include the temporal information in the state-action representation.
>
> References:
>
> [Dann et al., 2017] Christoph Dann, Tor Lattimore, Emma Brunskill, “Unifying PAC and Regret: Uniform PAC Bounds for Episodic Reinforcement Learning,” NeurIPS 2017.
>
> [Neu and Pike-Burke, 2020] Gergely Neu, Ciara Pike-Burke, “A Unifying View of Optimism in Episodic Reinforcement Learning,” NeurIPS 2020.

---

> ### Author Response · Authors · 2024-11-26
> **Response to Reviewer UwND**
>
> ### **Q6: How does BOFormer handle continuous domains in practice? Are there any discretization issues?**
> As BOFormer is a learning-based acquisition function (AF) implemented as a neural network, it can nicely handle continuous domains by leveraging the common ways of finding an (approximate) AF maximizer, just like the existing learning-based methods and neural AFs. Specifically:
> * Approximate maximization by Sobol grids (commonly used by learning-based methods): One can find an approximate maximizer of each learning-based AF by grid search with the help of Sobol sequences [Volpp et al., 2020, Hsieh et al., 2021]. Specifically, one can (i) first selects $N$ points spanning over the entire domain with the help of Sobol sequence, (ii) chooses the top-$M$ points in terms of AF value among the $N$ candidates, (iii) and then builds a local Sobol grid of $K$ points for each of these $M$ points. This is the default choice of BOFormer.
>
> * Gradient-based optimization with automatic differentiation (used by the rule-based methods in BoTorch):  One can also optimize the AF by using gradient-based optimization (e.g., multi-start L-BFGS-B in BOTorch https://botorch.org/docs/optimization), with the help of automatic differentiation provided by deep learning frameworks (e.g.,PyTorch). This is the default approach adopted by BoTorch.
>
> [Volpp et al., 2020] Meta-learning acquisition functions for transfer learning in bayesian optimization, Michael Volpp, Lukas P. Fröhlich, Kirsten Fischer, Andreas Doerr, Stefan Falkner, Frank Hutter, and Christian Daniel, International Conference on Learning Representation, 2020.
>
> [Hsieh et al., 2021] Reinforced few-shot acquisition function learning for Bayesian optimization, Bing-Jing Hsieh, Ping-Chun Hsieh, and Xi Liu. Neural Information Processing Systems, 2021
>
> ### **Q7: Provide the experimental details, e.g., (i) configuration of the surrogate model such as the choice of the kernel and hyperparameters and (ii) training cost for meta-training.**
> **(1) Configuration of the surrogate model**:
> - Training: For all the learning-based algorithms, we used the same GP surrogate model with an RBF kernel for obtaining the posterior distributions. Regarding the generation of GP synthetic functions as objective functions during training, each function is generated randomly under either an RBF or Matérn-5/2 kernel with lengthscale sampled from [0.1,0.4].
>
> - Testing: During testing, all algorithms utilized a surrogate model based on the Matérn-5/2 kernel. The length scale was estimated online by maximizing the marginal likelihood by following the same approach as BoTorch.
>
> **(2) Hyperparameters and implementation details of BOFormer**: We report the key hyperparameters and implementation details of BOFormer used in our experiments. Additional details of experiment can be found in Appendix A.2-A.3.
>
> **1. Transformer Architecture:**
> * Model: GPT-2-based Transformer architecture.
>     * Layers: 8 attention layers with 4 head.
>     * Hidden Size: 128 for all linear layers used to embed positional encodings, state-action pairs, rewards, and Q-values.
>
> **2. Q-Value Computation by target network for a Trajectory**
>
> Given a trajectory $\tau=\{ s_1, a_1, r_1, s_2, \cdots, s_{t-1}, a_{t-1},r_{t-1}\}$:
> * BOFormer computes $Q(h_0, s_1, a_1)$ using $(s_1, a_1)$ as the input.
> * Then, it computes $Q(h_1, s_2, a_2)$ by letting $(s_1, a_1, r_1, s_2,a_2)$ as the input.
> * This process continues sequentially until $Q(h_{t-2}, s_{t-1},a_{t-1})$ is computed.
>
> **3. Action selection by policy network at time $t$**
>
> To select $a_t$ at time $t$, BOFormer samples actions from a softmax policy where the logits correspond to Q-values: $\text{Pr}(a|h_{t-1}, s_t) = \frac{\text{exp}\bigg(Q(h_{t-1}, s_t,a)\bigg)}{\sum_{a'\in\mathcal{X}} \text{exp}\bigg(Q(h_{t-1}, s_t,a')\bigg)}$
>
> **4. Training details:**
>
> * **Demo policy:** Each trajectory is collected using the demo policy with a probability of 0.01; otherwise, it is collected by BOFormer itself.
> * **Batch sampling:** BOFormer samples a batch from the prioritized trajectory experience replay buffer, with each batch containing 16 trajectories.
> * **Optimization:** Gradient descent is performed on the policy network using the Adam optimizer with:
>     * Learning rate: $10^{-5}$
>     * Dropout rate: $0.1$
> * **Target network:** The target network is used to compute $\{Q(h_{i-1},s_i,a_i)\}_{i=1}^{t-1}$, and synchronized with the policy network every 5 episodes.
>
> **(3) Training cost for meta-training**: Among all the baselines, FSAF is a metaRL-based approach and is the only method that requires meta-training. Specifically, FSAF was trained for 500 episodes for the 2-objective setting and 300 episodes for 3-objective setting, respectively.

---

> ### Author Response · Authors · 2024-11-26
> **Response to Reviewer UwND**
>
> ### **Q8: Provide experimental results to support the idea of demo-policy-guided exploration.**
>
> To further investigate the connection between BOFormer’s performance and the choice of demo policy, we conducted additional experiments comparing qNEHVI (the original choice) and NSGA2 as demo policies, as well as a baseline where no demo policy is used. For both BOFormer (w/i qNEHVI) and BOFormer (w/i NSGA2), we apply the same settings as the training configuration in the original manuscript.
>
> The results are available at https://imgur.com/a/s47MuTQ
>
> Based on these results, we can observe that:
> - BOFormer (w/i qNEHVI) achieves favorable improvement in attained hypervolume than that without using a demo policy. This demonstrates the benefit of off-policy learning via a demo policy in the context of MOBO. By contrast, the improvement offered by the NSGA2-based demo policy appears minimal.
> - Recall that qNEHVI performs generally better than NSGA2 on these tasks (i.e., RBF, Matern, and BC) as shown in Table 1 of the original manuscript and the performance profiles in Figure 3. Intuitively, we would expect that the trajectories contributed by qNEHVI can better help BOFormer explore the regions with higher hypervolume. This intuition also resonates with the general understanding that RL performance can be correlated with the data quality (e.g., [Kumar et al., 2019]).
>
> [Kumar et al., 2019] Aviral Kumar, Justin Fu, Matthew Soh, George Tucker, Sergey Levine, “Stabilizing Off-Policy Q-Learning via Bootstrapping Error Reduction,” NeurIPS 2019.
>
> ### **Q9: Have you tried to compare the direct implementation of Generalized DQN you mentioned in Section 4.1 with your proposed BOFormer?**
> A9: Recall that the direct implementation in Section 4.1 is designed to implement Generalized DQN with the per-step observation that consists of two parts:
> (i) The posterior distributions of all $K$ objective functions at all the domain points;
> (ii) The best function values observed so far (denoted by $y^{(i)\*}$, $i=1,...,K$).
> Hence, the per-step observation is $\{ \{\mu^{(i)}(x), \sigma^{(i)}(x), y^{(i)\*}\}_{x\in \mathbb{X}, i\in [K]} \}$.
> As described in Section 4.1, this representation design is subject to two practical issues:
> - **Issue 1 -- Scalability issue in sequence length**: Under this design, the sequence length of the per-step observation would grow linearly with the number of domain points and pose a stringent requirement on training.
> - **Issue 2 -- Limited cross-domain transferability**: As this observation representation is domain-dependent, the learned model is tightly coupled with the training domain and has very limited transferability.
>
> We have tried this direct implementation, and we found that this design severely suffers from the scalability issue (i.e., Issue 1) and requires extremely long training time. Below we report the training time that we observed:
> - Even under a fairly small input domain of only 50 points (i.e., $|\mathbb{X}|=50$), we observe that 20 training iterations already take more than 36 hours of wall clock time to complete. **To finish at least 1000 training iterations (based on our training experience with BOFormer), this direct implementation would require about 75 days to complete one training run.**
> - Notably, the above training was already on a high-end GPU server with NVIDIA RTX 6000 Ada Generation GPUs and Intel Xeon Gold 5515+ CPU.
>
> The above issue will be more severe under an input domain with more domain points (i.e., larger $|\mathbb{X}|$). This manifests that this direct implementation suffers significantly from the scalability issue and is rather impractical. This also motivates the proposed BOFormer for substantiating the Generalized DQN framework.
>
> ### **Q10: Mathematical formulation issues and inconsistencies.**
> Thank you for catching these. We have carefully reviewed and updated the manuscript to address all mathematical formulation inconsistencies for clarity.

---

> > ### Comment · Reviewer_UwND · 2024-11-27
> >
> > I greatly appreciate the authors' detailed responses and additional experiments. Based on all the replies, I am willing to increase my score.
> >
> > I still have a few minor questions. Have you conducted any ablation studies to validate the effects of positional encoding and temporal information? (I am not requesting additional experiments given the current timeline, but am curious whether including this additional information truly leads to performance improvements.)
> > Regarding Q6, I believe using Sobol grids would be a limitation for high-dimensional tasks, and perhaps this could be mentioned in the paper.
> >
> > Finally, I hope the authors can include the details and additional explanations from these responses in the updated version of the paper.

---

> > > ### Author Response · Authors · 2024-11-30
> > > **Response to Reviewer UwND**
> > >
> > > Thank you for the reply and for acknowledging our rebuttal.
> > >
> > > - Regarding the ablation studies for validating the effects of positional encoding and temporal information, given the current timeline, we try our best to conduct a preliminary ablation study on BOFormer by comparing several variants of BOFormer models:
> > >
> > >     (1) **With Positional Encoding and Temporal Information:** This is the original BOFormer.
> > >
> > >     (2) **Without Temporal Information:** Temporal information is removed from BOFormer during training.
> > >
> > >     (3) **Without Positional Encoding nor Temporal Information:** Both components were excluded from BOFormer during training.
> > >
> > >     To better observe the training behavior, we collected all training data using the demo policy and monitored the evolution of the loss curve during training. The rest of the configuration, such as hyperparameters and neural network architecture, was consistent across all three variants.
> > >
> > >     The evolution of the training loss for the three variants can be found here:  [Training loss of BOFormer](https://imgur.com/a/aOo3UO2).
> > >
> > >     We observe both (2) and (3) appear to suffer from divergence in terms of training loss. These results appear to align with the requirements of episodic RL, where knowing the budget is important for estimating Q-values. Temporal information plays a vital role in the need of this budget-related representation, while positional encoding is essential for enabling the Transformer to recognize each position as corresponding to a single token. Together, these components are helpful to BOFormer’s stability and convergence during training.
> > >
> > > - Moreover, we have added the discussion about Sobol grids as well as the additional explanations and implementation details in the updated manuscript. Thank you for the helpful suggestions.
> > >
> > > Again, we thank the reviewer for all the detailed review and the efforts put into helping us improve our submission.

---

> > > > ### Comment · Reviewer_UwND · 2024-12-01
> > > >
> > > > Thanks for the additional ablation study, it seems positional encoding and temporal information will seriously influence the performance (and also I am not sure why loss will go up), I would suggest authors include an additional analysis/discussion in the future revised manuscript.
> > > >
> > > > Based on our discussion and your additional efforts, I am happy to further increase my score to 8.

---

> ### Author Response · Authors · 2024-12-04
> **Response to Reviewer UwND**
>
> Thank you once again for the valuable comments and for recognizing the contributions of this work. Regarding the state-action representation, as mentioned in our paper, it is not only history-dependent but also related to the remaining budget, as highlighted in various episodic RL studies. The loss increases when the representation is insufficient to map the corresponding state-action pairs to the target Q-values, especially as the training process progresses and encounters a wider variety of environments.

---

### Official Review · Reviewer_tuhd · 2024-11-02

**Soundness:** 3
**Presentation:** 2
**Contribution:** 3
**Rating:** 6
**Confidence:** 3

**Summary:**

This paper introduces BOFormer, a novel approach to multi-objective Bayesian optimization (MOBO) that leverages non-Markovian reinforcement learning. The authors' primary contribution is addressing the hypervolume identifiability issue in MOBO through a generalized DQN framework implemented via a Transformer architecture. The work introduces several practical enhancements, including Q-augmented observation representation, prioritized trajectory replay buffer, and demo-policy-guided exploration. The method is comprehensively evaluated on both synthetic functions and real-world hyperparameter optimization tasks for 3D Gaussian Splatting, demonstrating competitive performance against existing approaches. The authors also show promising transfer learning capabilities across different numbers of objective functions.

**Strengths:**

The authors have identified and thoroughly addressed a fundamental issue in learning-based MOBO approaches - the hypervolume identifiability problem. This represents a significant contribution to the field. The theoretical framework connecting non-Markovian RL to MOBO is rigorously developed and mathematically sound.

The experimental evaluation is comprehensive, comparing the method against both classical and learning-based baselines across diverse scenarios.

**Weaknesses:**

The contribution appears somewhat incremental relative to existing approaches like OptFormer and NAP. While the authors introduce novel elements, the core methodology builds heavily on established techniques.

The motivation for using Transformers in this context needs stronger justification. Given the small data regime typical in Bayesian optimization, the choice of a Transformer architecture, which typically requires substantial data for effective training, requires more thorough explanation.

The theoretical justification for computational intractability due to "curse of dimensionality" (line 244) requires more precise argumentation and supporting references.

**Questions:**

Could you provide details about the experimental methodology, specifically:
- How many random repetitions were performed to ensure statistical significance?
- Were the same network architectures used consistently across all experiments?
- What are the specific architectural details of the Transformer implementation?
- How was stability analysis conducted? ie, stability wrt different transformer models and also different initialization.

Would it be possible to present results on more challenging problems requiring significantly more than 100 samples? The current evaluation up to 100 samples may not fully reflect real-world BO scenarios.

The method's effectiveness with limited data (approximately 10 samples) is surprising given the typically large parameter count of Transformer architectures. Could you elaborate on how this is achieved?

How does the method's performance scale with increasing problem complexity and dimensionality?

Could you provide a more detailed justification for choosing a Transformer architecture in this small data regime compared to simpler alternatives?

---

> ### Author Response · Authors · 2024-11-26
> **Response to Reviewer tuhd**
>
> We greatly appreciate the reviewer for the insightful comments. Below, we address the questions raised by the reviewer by providing the point-to-point response. We hope the response could help the reviewer further recognize our contributions. Thank you.
>
> ### **Q1: The contribution appears somewhat incremental relative to existing approaches like OptFormer and NAP. While the authors introduce novel elements, the core methodology builds heavily on established techniques.**
>
> Thank you for the feedback. We highlight the differences and the innovative aspects of BOFormer compared to OptFormer and NAP as follows:
>
> **BOFormer vs OptFormer**: While both methods leverage Transformers in their design, there are two fundamental differences:
>
>  **(1) Application scope**: BOFormer is designed to serve as a general-purpose optimization solver for MO black-box optimization, whereas OptFormer was proposed specifically for single-objective HPO. This difference in scope also reflects why BOFormer is designed to exhibit favorable cross-domain capabilities.
>
>  **(2) Learning framework**: Motivated by the hypervolume identifiability issue, through BOFormer, we propose a non-Markovian RL framework built on the generalized deep Q-network to learn an AF that accounts for long-term effects. In contrast, OptFormer takes a supervised learning perspective by reinterpreting HPO as a language modeling task, utilizing offline single-objective HPO datasets and a Transformer model for sequence prediction.
>
> **BOFormer vs NAP**: While both BOFormer and NAP are learning-based BO methods, there are two salient differences:
>
>  **(1) Application scope**: Recall that NAP focuses on single-objective problems and jointly learns an acquisition function and a probabilistic predictive model (without using GP) based on some pre-collected source dataset. Accordingly, NAP is task-specific as it requires a source dataset at training and needs to take domain points as inputs at learning the predictive distribution. As a result, NAP requires re-training upon handling a new task.
> By contrast, BOFormer is designed as a general-purpose MOBO optimization solver with cross-domain capabilities (i.e., the size and the dimensionality of the input domains can be different between the training and testing). Hence, a BOFormer model can be directly applied to unseen testing functions of different input domains without re-training.
>
>  **(2) Algorithm perspective**: NAP is designed to solve end-to-end single-objective BO by simultaneously learning the surrogate model's accuracy and maximizing the cumulative reward. BOFormer, on the other hand, serves as a Transformer-based AF trained based on a Generalized DQN framework, which is specifically designed to address the hypervolume identifiability issue in MOBO.
>
> ### **Q2: Could you provide a more detailed justification for choosing a Transformer architecture in the small data regime of Bayesian optimization?**
> Thanks for raising this point. We would like to first clarify that using a Transformer model, which requires sufficient data during training, can still seamlessly address the small data regime of BO during evaluation. Specifically:
> - Regarding the small data regime in BO, given a black-box function during evaluation (i.e., the testing phase of BOFormer), the goal is to approach the maximum or minimum of thai black-box function using only a small number of samples (say, $T$ samples, where $T$ is small). In the context of BOFormer, this $T$ essentially corresponds to the episode length in the language of RL (e.g., like in the classic Cartpole environment in OpenAI Gym, the episode length is $200$ steps and hence $T=200$). As we can set a small $T$ (e.g., $T=100$ in our configuration) during BOFormer training, the learned BOFormer can tackle the inherent small-data issue in the general BO.
>
> - On the other hand, the training of BOFormer does require sufficient training episodes (say $K$ episodes), each of which correspond to taking $T$ samples of a set of $d$ synthetic GP black-box functions ($d$ denotes the number of objectives in MOBO). In practice, we find that setting $K=3000$ is a good choice with favorable convergence behavior. As BOFormer is trained solely on synthetic GP functions, the cost of generating these training data is actually low and hence fairly acceptable.
>
> Remark: The above description also addresses the question Q7 below.
>
> ### **Q3: The motivation for using a Transformer architecture**
> Recall that to address the non-Markovian nature of the MOBO problem, we present generalized DQN, a non-Markovian RL framework, and implement this by employing sequence modeling. While there are multiple viable options of sequence modeling (e.g., Transformers and RNNs), we chose Transformers due to their ability to effectively model long-term dependencies and capture complex relationships in sequential data, which are critical in this setting.

---

> ### Author Response · Authors · 2024-11-26
> **Response to Reviewer tuhd**
>
> ### **Q4: The theoretical justification for computational intractability due to curse of dimensionality.**
> By formulating BO as an MDP for multi-step lookahead optimization, the intractability arises from several key factors:
> * In continuous domains, the state and action spaces are uncountably large
> * Extending the decision-making horizon to multiple steps causes the number of potential outcomes to grow exponentially
> * Multi-step lookahead policies require recursive computations involving both maximization and integration over continuous domains. These characteristics make BO analytically intractable, except in simplified cases.
>
> Several prior works (e.g., [Jiang et al., 2020a; Paulson et al., 2022; Jiang et al., 2020b]) have also discussed the intractability of deriving optimal policies in BO. These works underscore the inherent difficulties of multi-step optimization in BO and the need for approximations to tackle its intractability.
>
> [Jiang et al., 2020a] Shali Jiang, Daniel Jiang, Maximilian Balandat, Brian Karrer, and Jacob Gardner, and Roman Garnett, "Efficient Nonmyopic Bayesian Optimization via One-Shot Multi-Step Trees," NeurIPS 2020.
>
> [Paulson et al., 2022] Joel A. Paulson, Farshud Sorouifar, and Ankush Chakrabarty, "Efficient multi-step lookahead Bayesian optimization with local search constraints," CDC 2022.
>
> [Jiang et al., 2020b] Shali Jiang, Henry Chai, Javier Gonzalez, and Roman Garnett, "BINOCULARS for Efficient, Nonmyopic Sequential Experimental Design," ICML 2020.
>
> ### **Q5: Provide more details about the experimental methodology, including number of random repetitions and stability, consistency of the network architectures used, and specific architectural details of the Transformer**
>
> * **Random repetitions and stability:**
> To analyze the stability of BOFormer, we report the results of three BOFormer models trained under different random seeds (and hence different initializations). For a fair comparison, all the three models are trained for 1000 iterations. The results of final averaged regrets are shown in the table below.
>
>     |       | BOFormer (seed #1) | BOFormer (seed #2)  | BOFormer (seed #3)  |
>     | ----- | -------  | ------- | ------- |
>     | RBF	| 0.83527  | 0.83414 | 0.83402 |
>     | matern| 0.85999  | 0.85907 | 0.85766 |
>     | BC    | 0.41951  | 0.41882 | 0.41955 |
>     | DRa    | 0.92234  | 0.92208 | 0.92216 |
>
>     We observe that the performance across these BOFormer models remains nearly identical, indicating that BOFormer is a relaible and stable algorithm and unaffected significantly by the variations in training seed and initialization.
>
> * **Consistency of the network architectures used and specific architectural details**:
> The same network architecture was used consistently across all the experiments for BOFormer. Specifically, we leverage the GPT-2 implementation (https://github.com/huggingface/transformers/tree/main/src/transformers/models/gpt2) provided by the Hugging Face Transformers library. Additional architectural details of the Transformer implementation are provided in Appendix A.2 of the original manuscript.
>
> ### **Q6: The performance of BOFormer on more challenging problems with more than 100 samples**
> Thanks for the helpful suggestion. To address this, we conducted additional experiments on various challenging problems with more than 100 samples, including scenarios with (1) higher domain dimensionality and (2) increased perturbation noise. Accordingly, we set $T=200$ or all the following experiments to better evaluate BOFormer and other baselines.
>
> The results demonstrate that BOFormer remains competitive compared to other MOBO baseline methods across various types of challenging scenarios. These findings highlight BOFormer’s scalability, making it well-suited for more challenging black-box functions.
>
> - DTLZ (d=100): https://imgur.com/a/YMj2sZz
> - HPO-3DGS : https://imgur.com/a/nerf-t200-8l4cihz
>
> Remark 1: DTLZ is a family of synthetic functions introduced in the [pymoo library](https://pymoo.org/).
>
> Remark 2: To make the problems more challenging as suggested, for the HPO-3DGS tasks, the perturbation noise is set as 0.1 to make the black-box functions more non-smooth.
>
> Remark 3: In the above, we compare BOFormer with the rule-based methods such as qNEHVI, JES, and qParEGO as well as the learning-based method like FSAF. These MOBO methods have been shown to be the most competitive among all the baselines (cf. Tables 1-2 in the original manuscript)
>
> Remark 4: FSAF is a metaRL-based method and requires some metadata for few-shot adaptation. By contrast, BOFormer is evaluated on unseen testing functions in a zero-shot manner.

---

> ### Author Response · Authors · 2024-11-26
> **Response to Reviewer tuhd**
>
> ### **Q7: Scalability of BOFormer to high-dimensional problems**
>
> To address the concerns about the scalability to high-dimensional problems, we conducted additional experiments on black-box functions with higher-dimensional domains, including:
>
> - (Ackley, Rosenbrock) and (Ackley & Rastrigin) (d=40): https://imgur.com/a/ar-ara-dim40-F8KrQ3z
> - Matern52 and RBF (d=10 and 30): https://imgur.com/a/lbgyIuS
> - DTLZ (d=100): https://imgur.com/a/YMj2sZz
>
> The above results demonstrate that BOFormer remains competitive compared to other MOBO baseline algorithms on these high-dimensional tasks. while FSAF requires metadata for few-shot updates,. Notably, all the above results are obtained under the same BOFormer model without any fine-tuning, and hence this further demonstrates the strong cross-domain transferability of BOFormer.
>
> ### **Q8: The method's effectiveness with limited data (approximately 10 samples) is surprising given the typically large parameter count of Transformer architectures. Could you elaborate on how this is achieved?**
> As described in the response to Q2, the issue with limited data during evaluation (i.e., small data regime) essentially corresponds to a small episode length in the context of RL.
> As we can set a small $T$ (e.g., $T$=100 in our configuration) during BOFormer training, the learned BOFormer can tackle the inherent small-data issue and attain a good hypervolume with only a small number of samples.
>
> ### **Q9: How does the method's performance scale with increasing problem complexity and dimensionality?**
> Based on our response to Q6 and Q7, through the additional experiments, we can see that BoFormer can indeed scale to problerms of increased complexity and higher domain dimensionality.

---

### Official Review · Reviewer_62Xe · 2024-11-04

**Soundness:** 3
**Presentation:** 3
**Contribution:** 3
**Rating:** 6
**Confidence:** 3

**Summary:**

This paper introduces BOFormer, a novel learning-based acquisition function for multi-objective Bayesian optimization (MOBO) that combines reinforcement learning and sequence modeling with Transformers. BOFormer addresses the hypervolume identifiability issue in MOBO, which stems from its non-Markovian nature, by presenting a Generalized DQN framework and substantiating it through sequence modeling.
Several practical enhancements, such as Q-augmented observation representation and prioritized trajectory replay buffer, are incorporated to facilitate the training of BOFormer. Extensive experiments on synthetic and real-world hyperparameter optimization problems demonstrate that BOFormer consistently outperforms various benchmark methods, exhibits cross-domain transfer capabilities, and efficiently transfers learning across different numbers of objective functions.

**Strengths:**

One of this paper's key strengths is its novel approach to addressing the hypervolume identifiability issue in multi-objective Bayesian optimization (MOBO). By presenting the Generalized DQN framework and implementing it through BOFormer, the authors tackle MOBO's inherent non-Markovian nature. This innovative perspective of reinterpreting MOBO as a sequence modeling problem using Transformers allows for a more effective and efficient solution to the identifiability issue.
Another strength lies in the practical enhancements introduced to facilitate the training of BOFormer. The Q-augmented observation representation provides an informative indicator of the prospective improvement in hypervolume while maintaining a domain-agnostic and memory-efficient representation. The prioritized trajectory replay buffer and off-policy learning enable better convergence and data efficiency during training. Furthermore, the demo-policy-guided exploration ensures efficient search space exploration, which is particularly important given the limited sampling budget in MOBO.

**Weaknesses:**

Limited theoretical analysis: Although the paper introduces the Generalized DQN framework and provides empirical evidence of its effectiveness, it lacks an in-depth theoretical analysis of the proposed approach. A more rigorous theoretical foundation could help better understand BOFormer's convergence properties, optimality guarantees, and limitations.

Scalability to high-dimensional problems: While BOFormer performs well on the tested problems, its scalability to high-dimensional MOBO problems is not explicitly addressed. As the number of dimensions increases, the search space grows exponentially, potentially impacting the algorithm's performance. Further investigation into the scalability of BOFormer and performance on high-dimensional problems would be valuable.

**Questions:**

How does the Generalized DQN framework differ from the standard DQN in terms of theoretical guarantees and convergence properties? Can the authors provide more insights into the theoretical foundations of their approach?

The paper introduces several practical enhancements, such as Q-augmented observation representation and prioritized trajectory replay buffer. How do these enhancements contribute to the performance of BOFormer individually, and are there any potential synergies or trade-offs between them?

The demo-policy-guided exploration is an interesting concept. How sensitive is BOFormer's performance to the choice of the demo policy, and what are the characteristics of a good demo policy for MOBO problems?

The paper showcases BOFormer's cross-domain transfer capabilities and efficient transfer learning across different numbers of objective functions. Can the authors elaborate on the potential limitations of this transfer learning approach and discuss scenarios where it might not be applicable or effective?

---

> ### Author Response · Authors · 2024-11-26
> **Response to Reviewer 62Xe**
>
> We sincerely thank the reviewer for the insightful feedback on this work. Below, we address the questions raised by the reviewer by providing the point-to-point response. We hope the response could help the reviewer further recognize our contributions. Thank you.
>
> ### **Q1:  Provide more insights into the theoretical foundations of Generalized DQN**
>
> Thank you for the helpful suggestion. The design of Generalized DQN is theoretically grounded on the generalized version of Bellman optimality equations presented in Proposition 1. Specifically:
> Equations (2) and (3) uniquely characterize the optimal Q function and value function (denoted by $Q^*$ and $V^*$) for non-Markovian RL. Under Generalized DQN, we would like to learn $Q^*$ under function approximation, i.e., learn a parametric function $Q_{\theta}$ that matches $Q^*$ either exactly or approximately.
> Under Generalized DQN, this is achieved by learning $Q_{\theta}$ that minimizes the following loss function,
>
> \begin{align}
> \mathcal{L}(Q_\theta)&=\mathbb{E}_{(h,a,o)\sim \mathcal{D}}\bigg[\Big(r(h,a,o)+\gamma \max\_{a'\in \mathcal{A}} Q\_{\theta}(h',a')-Q\_{\theta}(h,a)\Big)^2\bigg].
> \end{align}
>
> Then, there are two properties that we can obtain:
>
> **Property 1**: If there exists a $\theta^*$ with $Q_{\theta^*}(h,a)=Q^*(h,a)$ for all $(h,a)$ pair, then $Q_{\theta^*}$ would have zero loss, i.e., $\mathcal{L}(Q_{\theta^*})=0$, regardless of the data distribution $\mathcal{D}$. That is, $\theta^*$ is a minimizer of the loss function. Note that Property 1 can be directly verified by plugging Equations (2) and (3) into the above loss function.
>
> To ensure that the loss can be effectively minimized, one shall use a parametric model that has sufficiently good representation power, such as Transformers that can serve as universal function approximators for sequence-to-sequence problems [Yun et al., 2020].
>
> **Property 2**: If one can find a $\theta^\dagger$ with zero loss $\mathcal{L}(Q_{\theta^\dagger})=0$, then we have $Q_{\theta^\dagger}(h,a)=Q^*(h,a)$ for all $(h,a)$ pairs with non-zero sampling probability. This again can be directly verified by using Equations (2) and (3).
>
> Based on these two properties, we know that under a sufficiently powerful model (e.g., Transformer) and a good data coverage (e.g., through exploration), minimizing the loss function of Generalized DQN can learn the optimal Q function $Q^*$.
>
> [Yun et al., 2020] Chulhee Yun, Srinadh Bhojanapalli, Ankit Singh Rawat, Sashank J. Reddi, and Sanjiv Kumar, “Are Transformers universal approximators of sequence-to-sequence functions?” ICLR 2020.
>
>
> ### **Q2: Scalability of BOFormer to high-dimensional problems**
> Thank you for raising this important point. To address the concerns about the scalability to high-dimensional problems, we conducted additional experiments on black-box functions with higher-dimensional domains, including:
>
> - (Ackley, Rosenbrock) and (Ackley & Rastrigin) (d=40): https://imgur.com/a/ar-ara-dim40-F8KrQ3z
> - Matern52 and RBF (d=10 and 30):https://imgur.com/a/lbgyIuS
> - DTLZ (d=100): https://imgur.com/a/YMj2sZz
>
> Remark 1: DTLZ is a family of synthetic functions originally introduced by [Deb et al., 2005] for MOBO and included in the [pymoo library](https://pymoo.org/).
>
> Remark 2: In the above, we compare BOFormer with the rule-based methods such as qNEHVI, JES, and qParEGO as well as the learning-based method like FSAF. These MOBO methods have been shown to be the most competitive among all the baselines (cf. Tables 1-2 in the original manuscript)
>
> Remark 3: FSAF is a metaRL-based method and requires some metadata for few-shot adaptation. By contrast, BOFormer is evaluated on unseen testing functions in a zero-shot manner.
>
> The above results demonstrate that BOFormer remains competitive compared to other MOBO baseline algorithms on these high-dimensional tasks. while FSAF requires metadata for few-shot updates,. Notably, all the above results are obtained under the same BOFormer model without any fine-tuning, and hence this further demonstrates the strong cross-domain transferability of BOFormer.
>
> [Deb et al., 2005] Kalyanmoy Deb, Lothar Thiele, Marco Laumanns, and Eckart Zitzler, "Scalable test problems for evolutionary multiobjective optimization," Evolutionary Multiobjective oOptimization: Theoretical Advances and Applications, 2005.

---

> ### Author Response · Authors · 2024-11-26
> **Response to Reviewer 62Xe**
>
> ### **Q3: Ablation study and intuition of practical components in BOFormer: Q-augmented representation and prioritized trajectory replay buffer**
> Recall that the enhancements introduced in BOFormer, such as the Prioritized Trajectory Replay Buffer and Q-augmented observation representation, contribute to its performance in the following ways:
> * **Prioritized Trajectory Experience Replay Buffer (PTER)** ensures that the most informative transitions (e.g., those with significant temporal-difference errors) are replayed more frequently during training, accelerating convergence.
> * **Q-Augmented Observation Representation**: The Q-augmented representation acts as an informative indicator of the potential improvement in hypervolume. By incorporating this design, BOFormer can also be interpreted as a sequence prediction method, akin to next-word prediction in natural language processing. This perspective helps BOFormer efficiently model and predict future improvements in the optimization process.
>
> We further conducted an ablation study on these two components, namely Q-augmented representation and prioritized trajectory replay buffer,  utilized in BOFormer.
>
> The results are available at https://imgur.com/a/gKETZRY
>
> The above results demonstrate that both components contribute significantly to improving the performance of BOFormer.
>
> ### **Q4: The demo-policy-guided exploration is an interesting concept. How sensitive is BOFormer's performance to the choice of the demo policy, and what are the characteristics of a good demo policy for MOBO problems?**
>
> To further investigate the connection between BOFormer’s performance and the choice of demo policy, we conducted additional experiments comparing qNEHVI (the original choice) and NSGA2 as demo policies, as well as a baseline where no demo policy is used. For both BOFormer (w/i qNEHVI) and BOFormer (w/i NSGA2), we apply the same settings as the training configuration in the original manuscript.
>
> The results are available at https://imgur.com/a/s47MuTQ
>
> Based on these results, we can observe that:
> - BOFormer (w/i qNEHVI) achieves favorable improvement in attained hypervolume than that without using a demo policy. This demonstrates the benefit of off-policy learning via a demo policy in the context of MOBO. By contrast, the improvement offered by the NSGA2-based demo policy appears minimal.
> - Recall that qNEHVI performs generally better than NSGA2 on these tasks (i.e., RBF, Matern, and BC) as shown in Table 1 of the original manuscript and the performance profiles in FIgure 3. Intuitively, we would expect that the trajectories contributed by qNEHVI can better help BOFormer explore the regions that can attain higher hypervolume. This intuition also resonates with the general understanding that RL performance can be correlated with the data quality [Kumar et al., 2019].
>
> [Kumar et al., 2019] Aviral Kumar, Justin Fu, Matthew Soh, George Tucker, Sergey Levine, “Stabilizing Off-Policy Q-Learning via Bootstrapping Error Reduction,” NeurIPS 2019.
>
> ### **Q5: The paper showcases BOFormer's cross-domain transfer capabilities and efficient transfer learning across different numbers of objective functions. Can the authors elaborate on the potential limitations of this transfer learning approach and discuss scenarios where it might not be applicable or effective?**
>
> Thank you for the insightful question. In the general context of transfer learning, the effectiveness of cross-domain transfer is known to depend on the similarity between the domains. In MOBO, the domain similarity can be characterized from various aspects, such as the domain dimensionality, number of objectives, and smoothness of the black-box functions. Through the experiments in the original manuscript and in our response, we can see that BO can nicely adapt to various dimensionalities and functions of various levels of smoothness, and can achieve transfer the 2-objective model to a 3-objective BOFormer via fine-tuning.
>
> To further explore the limit of transferability, we conduct additional experiments by training a 4-objective BOFormer model fine-tuned from a 2-objective model and test it on 4-objective tasks with functions of higher domain dimensionality (d=40).
>
> The results are available at https://imgur.com/a/mtn5ozB
>
> In such cases, we observe that BOFormer still achieves a hypervolume comparable to other MOBO baselines throughout different sampling stages, though not necessarily being the highest one. These findings corroborate that the effectiveness of BOFormer's transfer learning does depend on the domain dissimilarity.

---

### Official Review · Reviewer_91dP · 2024-11-06

**Soundness:** 3
**Presentation:** 2
**Contribution:** 2
**Rating:** 5
**Confidence:** 3

**Summary:**

The paper introduces BOFormer, a novel approach for tackling multi-objective Bayesian optimization (MOBO) challenges, particularly the non-Markovian nature and hypervolume identifiability issues. BOFormer employs non-Markovian reinforcement learning and sequence modeling, leveraging the Transformer architecture to optimize long-term outcomes. The method demonstrates superior performance in various synthetic and real-world multi-objective optimization scenarios.

**Strengths:**

The paper introduces an innovative approach by framing multi-objective Bayesian optimization (MOBO) as a non-Markovian reinforcement learning problem. This represents a creative combination of existing ideas from non-Markovian RL and Transformer-based sequence modeling, marking a fresh perspective in the field.
The use of diagrams and examples, such as the hypervolume identifiability issue, aids in understanding complex concepts.

**Weaknesses:**

1. The discussion of shortcomings in the paper is relatively brief and does not clearly articulate the innovative aspects of the work. Furthermore, the contributions appear to be somewhat incremental rather than groundbreaking.
2. The use of Transformers may require substantial computational resources and memory, which could limit accessibility for some users.
3. The explanations regarding the experimental section lack clarity. The paper does not specify how the proposed algorithm's time efficiency compares to that of other algorithms. Additionally, it is unclear whether the non-Markovian nature of the process consumes more time than a Markovian approach. The improvements over baseline results also do not appear to be significant.

**Questions:**

1. How does the time efficiency of your proposed algorithm compare to the baseline algorithms?
2. What is the intended meaning of the experimental results in your Figure 3?

---

> ### Author Response · Authors · 2024-11-25
> **Response to Reviewer 91dP**
>
> We greatly appreciate the reviewer for the insightful comments. Below, we address the questions raised by the reviewer by providing the point-to-point response. We hope the response could help the reviewer further recognize our contributions. Thank you.
>
> ### **Q1: Discussion on contributions, innovative aspects, and potential shortcomings**
>
> We highlight the innovative aspects of this paper as follows:
>
> **1. A novel research perspective on MOBO**: We identify the hypervolume identifiability issue in MOBO, which arises due to the inherent non-Markovianty in MOBO, as the improvement in hypervolume is history-dependent. This issue presents a fundamental challenge in MOBO as it complicates accurate predictions of hypervolume improvement. This insight thereby motivates a new research question about how to learn a non-myopic acquisition function for MOBO without suffering from this identifiability issue.
>
> **2. A new algorithm based on Generalized DQN**: To address the inherent non-Markovianty in MOBO, we propose a novel algorithm based on Generalized DQN with sequential modeling. This approach enables the agent to learn under history-dependent reward functions through a Non-Markovian RL framework, effectively addressing the hypervolume identifiability issue.
>
> **3. Novelty in orchestrating modules for a practical implementation of Generalized DQN**:
> To tackle the complexities of training and inference in Non-Markovian RL, we propose BOFormer with several key designs:
>
> (i) Transformer for sequential modeling: We leverage Transformers due to their ability to effectively model long-term dependencies and capture complex relationships in sequential data, which are critical in the MOBO setting.
>
> (ii) Q-Augmented observation representation: The Q-augmented representation provides an informative indicator of potential hypervolume improvement. Under this design, BOFormer can be interpreted as a sequence prediction method, similar to next-word prediction in language modeling.
>
> (iii) Prioritized trajectory replay buffer: This ensures that the most informative trajectories (e.g., those with large temporal-difference errors) are replayed more frequently during training, accelerating convergence and improving learning efficiency.
>
> Notably, despite that some of the above techniques are not completely new by itself, their integration for implementing Generalized DQN is indeed new.
>
> On the other hand, just like all the other learning-based BO methods, one shortcoming of BOFormer is the need for training, which is typically not needed by rule-based methods. That being said, we would like to pinpoint that the training overhead of BOFormer is fairly mild for three reasons:
>
> (i) Recall that BOFormer serves as a general-purpose MOBO optimization solver with superior cross-domain capability (i.e., the size and the dimensionality of the input domains can be different between the training and testing). As a result, while training is required by BOFormer, the training cost can be essentially amortized as one BOFormer model can be repeatedly applied to optimizing a large variety of black-box functions in a zero-shot manner.
>
> (ii) The proposed BOFormer is trained solely on synthetic GP functions, which are relatively cheap to generate (compared to real-world functions), and then can be deployed to optimize unseen testing functions.
>
> (iii) The training of BOFormer can be done on commercial GPUs (e.g., RTX 40 series), and the computational requirement is fairly acceptable (please see Q2 for more details).

---

> ### Author Response · Authors · 2024-11-26
> **Response to Reviewer 91dP**
>
> ### **Q2: Comparison of computational resources and inference time between BOFormer and baselines**
>
> To answer this, we report the maximum GPU memory usage and the average per-sample inference time as follows:
>
> - **Inference time**: The results of per-sample inference time (in seconds) of BOFormer and the baseline algorithms are provided as follows. To ensure a fair comparison, we evaluate the inference time of all the algorithms on the same computing infrastructure.
>     | Algorithms | 2 Obj. | 3 Obj. | 4 Obj. |
>     |:----------:|:------:|:------:|:------:|
>     |  BOFormer  | 0.1420 | 0.1420 | 0.1429 |
>     |   qNEHVI   | 0.0615 | 0.8042 | 2.2410 |
>     |    qHVKG   | 3.7166 | 5.3337 | 6.6276 |
>     |     JES    | 0.4666 | 0.5015 | 0.5787 |
>     |    NSGA2   | 0.0017 | 0.0019 | 0.0020 |
>     |   qParEGO  | 0.0110 | 0.0142 | 0.0175 |
>     |    FSAF    | 0.0549 | 0.0655 | 0.0692 |
>     |  OptFormer | 2.3733 | 3.6615 | 4.8551 |
>     |     DT     | 0.0130 | 0.0128 | 0.0128 |
>     |     QT     | 0.0937 | 0.0971 | 0.0973 |
>
>
>     The messages are mainly three-fold:
>     - **BOFormer vs SOTA rule-based methods (e.g., qNEHVI, qHVKG, and JES)**: We observe that for the baselines that are competitive in the attained hypervolume (such as qNEHVI and qHVKG), the inference time scales significantly with the number of objectives, whereas BOFormer maintains a lower and nearly constant inference time across different number of objectives.
>     - **BOFormer (non-Markovia) vs FSAF (Markovian)**: Compared to the Markovian approach like FSAF, BOFormer only needs a slightly higher per-step inference time and remains rather efficient. Hence, the non-Markovian nature of BOFormer still consumes an inference time comparable to a Markovian approach.
>     - **BOFormer vs OptFormer**: BOFormer enjoys a much lower inference time than the sophisticated OptFormer, which views black-box optimization as a language modeling problem in a text-to-text manner and hence involves text-based input representation.
>
> - **Memory usage**:
>     To address the concerns about resource requirements, we include the memory usage of the learning-based algorithms as follows.
>
>     |           | Training| Testing |
>     | --------- | ------- | ------- |
>     | BOFormer  | 3642 MB | 3596 MB |
>     | OptFormer | 8758 MB | 1446 MB |
>     | FSAF      | 672 MB  | 294 MB  |
>     | DT        | 1334 MB | 672 MB  |
>     | QT        | 2092 MB | 2046 MB |
>
>     While BOFormer utilizes a Transformer-based model, its memory usage during both training and inference remains manageable and is well-suited for most commercial GPUs, such as the NVIDIA RTX 40 series.
>
> ### **Q3: More explanation on the performance profiles in Figure 3**
>
> The performance profiles in Figure 3 are meant to more reliably present the performance variability of BOFormer and other baselines across testing episodes than the interval estimates of aggregate metrics. Specifically, the performance profiles show the (empirical) score distributions, indicating the fraction of runs that achieve a score above a certain normalized threshold. The performance profile was originally introduced by [Dolan and Moré, 2002] for benchmarking optimization software and later recommended by [Agarwal et al., 2021] to deep RL.
>
> In the context of MOBO, Figure 3 shows the score distribution of the attained final hypervolume, and the more top-right a curve resides the better.
> Based on Figure 3, we observe that in most tasks, the profiles of BOFormer (i.e., the curves in red) sit on the top right of the other MOBO baselines and hence enjoy a statistically better performance in terms of hypervolume.
>
> In summary, Figure 3 shows that the improvements of BOFormer over baselines are indeed significant.
>
> [Agarwal et al., 2021] R. Agarwal, M. Schwarzer, P. S. Castro, A. C. Courville, and M. Bellemare, "Deep reinforcement learning at the edge of the statistical precipice," NeurIPS 2021.
>
> [Dolan and Moré, 2002] Elizabeth D Dolan and Jorge J Moré, “Benchmarking optimization software with performance profiles,” Mathematical Programming, 2002.

---

### Author Response · Authors · 2024-11-30
**Response to All Reviewers**

We thank the reviewers for their insightful feedback and constructive suggestions. We have updated our manuscript to address the points raised, with modifications highlighted in blue. Below, we further highlight the responses to the key concerns.

## **(1) Additional experimental results**
* **Scalability of BOFormer to high-dimensional problems**

   We conducted additional experiments on black-box functions with higher-dimensional domains, including:

    - Matern52 and RBF (d=10 and 30): https://imgur.com/a/lbgyIuS
    - (Ackley, Rosenbrock) and (Ackley & Rastrigin) (d=40): https://imgur.com/a/ar-ara-dim40-F8KrQ3z
    - DTLZ (d=100): https://imgur.com/a/YMj2sZz

    The above results demonstrate that BOFormer remains competitive compared to other MOBO baseline algorithms on these high-dimensional tasks. Notably, all the above results are obtained under the same BOFormer model without any fine-tuning, and hence this further demonstrates the strong cross-domain transferability of BOFormer.

    **Remark 1**: DTLZ is a family of synthetic functions for MOBO and included in the [pymoo library](https://pymoo.org/).

    **Remark 2**: In the above, we compare BOFormer with the rule-based methods such as qNEHVI, JES, and qParEGO as well as the learning-based method like FSAF. These MOBO methods have been shown to be the most competitive among all the baselines (cf. Tables 1-2 in the original manuscript). Moreover, FSAF is a metaRL-based method and requires some metadata for few-shot adaptation. By contrast, BOFormer is evaluated on unseen testing functions in a zero-shot manner.

* **The performance of BOFormer on more challenging problems with more than 100 samples**

    We conducted additional experiments on various challenging problems with more than 100 samples, including scenarios with (1) higher domain dimensionality and (2) increased perturbation noise. Accordingly, we set $T=200$ or all the following experiments to better evaluate BOFormer and other baselines.

    - DTLZ (d=100): https://imgur.com/a/YMj2sZz
    - HPO-3DGS : https://imgur.com/a/nerf-t200-8l4cihz

    The results demonstrate that BOFormer remains competitive compared to other MOBO baseline methods across various types of challenging scenarios. These findings highlight BOFormer’s scalability, making it well-suited for more challenging black-box functions.

    **Remark 3**: To make the problems more challenging for the HPO-3DGS tasks, the perturbation noise is set as 0.1 to make the black-box functions more non-smooth.


## **(2) Ablation study of demo policy**
We conducted additional experiments comparing qNEHVI (the original choice) and NSGA2 as demo policies, as well as a baseline where no demo policy is used. For both BOFormer (w/i qNEHVI) and BOFormer (w/i NSGA2), we apply the same settings as the training configuration in the original manuscript.

The results are available at https://imgur.com/a/s47MuTQ

Key Observations:
- BOFormer (w/i qNEHVI) achieves favorable improvement in attained hypervolume than that without using a demo policy. This demonstrates the benefit of off-policy learning via a demo policy in the context of MOBO. By contrast, the improvement offered by the NSGA2-based demo policy appears minimal.
- Recall that qNEHVI performs generally better than NSGA2 on these tasks (i.e., RBF, Matern, and BC) as shown in Table 1 of the original manuscript and the performance profiles in Figure 3. Intuitively, we would expect that the trajectories contributed by qNEHVI can better help BOFormer explore the regions with higher hypervolume. This intuition also resonates with the general understanding that RL performance can be correlated with the data quality (e.g., [Kumar et al., 2019]).

## **(3) More explanation on the performance profiles in Figure 3**

The performance profiles in Figure 3 are meant to more reliably present the performance variability of BOFormer and other baselines across testing episodes than the interval estimates of aggregate metrics. Specifically, the performance profiles show the (empirical) score distributions, indicating the fraction of runs that achieve a score above a certain normalized threshold. The performance profile was originally introduced by [Dolan and Moré, 2002] for benchmarking optimization software and later recommended by [Agarwal et al., 2021] to deep RL.

In the context of MOBO, Figure 3 shows the score distribution of the attained final hypervolume, and the more top-right a curve resides the better.
Based on Figure 3, we observe that in most tasks, the profiles of BOFormer (i.e., the curves in red) sit on the top right of the other MOBO baselines and hence enjoy a statistically better performance in terms of hypervolume.

In summary, Figure 3 shows that the improvements of BOFormer over baselines are indeed significant.

---

### Meta-Review · Area_Chair_YWda · 2024-12-24

**Metareview:**

This paper frames multi-objective Bayesian optimization as a reinforcement learning problem.  They use deep Q-learning with a Transformer architecture to model the value function and use normalized hypervolume improvement as the reward.  This framing allows the authors to take the entire sequence of evaluations into account when proposing the next experiment to run.  The authors argue that this is an improvement over standard myopic procedures because it addresses what they call the 'hypervolume identifiability issue'.  Specifically, the same hypervolume improvement can be achieved in multiple ways which can't be distinguished with a myopic approach.  Thus knowing what has been attempted before is important to make the search more efficient.  The authors show that there method is effective across a variety of synthetic benchmarks and they construct an optimization benchmark on 3d object reconstruction problems.

The reviews are leaning positive with one leaning reject, two leaning accept and one accept (5, 6, 6, 8).  Thus the average score is a marginal accept.  Multiple reviewers found the paper somewhat incremental - i.e. existing literature has created RL acquisition functions based on Q-learning.   Although they noted that this paper does extend the setting to multi-objective and as including the sequence ("non-Markovian").  The reviewers also noted that there was not much theoretical justification for the proposed approach - e.g. there doesn't seem to be any understanding of convergence guarantees.  Reviewers also requested more details regarding the experimental setup (the authors appear to have added a significant amount in the rebuttal).   On the positive side, the reviewers seemed to find the problem setting compelling and the non-identifiability argument interesting and compelling.  They seemed to find the experiments comprehensive and convincing.

In my own reading of the paper, I found the paper somewhat unclear and difficult to follow.  The method doesn't really seem like Bayesian optimization in my opinion, i.e. it isn't really Bayesian AFAICT.  Instead, it seems more like using deep Q-learning with transformers for multi-objective optimization.  That said, the approach does seem novel, albeit perhaps somewhat incremental over existing Q-learning for single objective optimization approaches.  The identifiability issue is an interesting observation, and reviewers also found this insightful.  The experiments seem positive, i.e. the proposed approach is competitive or better than the baselines in general.  However, not all of the plots seem that convincing - e.g. in figure 3 BOFormer doesn't seem consistently better than other methods.  Also, it's customary to show error bars on plots for Bayesian optimization given the variability in performance of methods across different random restarts.  The authors state that they present average results, but it's unclear to me how many runs were averaged over.  Overall, the problem is well motivated, the approach is novel and there does seem to be merit in the approach and the ideas presented in the work.  In my own opinion, I don't necessarily object strongly to acceptance but I think the paper could be made stronger through some careful editing, revisions and by providing error bars on experiments.  In fact I think the lack of error-bars can be sufficient reason to reject - we don't know if the improvements are statistically significant and I would expect the results to be high-variance given the method (and optimization trajectories in general).

**Additional Comments On Reviewer Discussion:**

The authors provided detailed responses to all of the reviews including details of computational costs, additional experimental results, lots of additional details about the experiments and ablation studies.  Unfortunately, only one reviewer seems to have responded to the author response and raised their score to an 8.  Given that the other reviewers did not acknowledge the author response, and the only reviewer that did raised their score to accept, this makes me lean more towards accept.  I.e. the average review is leaning accept and most reviewers did not take the author response into account.  Given that the response is significant and seems to comprehensively address the reviewers concerns, it seems reasonable to give the authors the benefit of the doubt here.

---

### Decision · Program_Chairs · 2025-01-22

Accept (Poster)